# Quantifying Distributional Invariance in Causal Subgraph for IRM-Free Graph Generalization

Yang Qiu[1], Yixiong Zou[1*], Jun Wang[2*], Wei Liu[1], Xiangyu Fu[1], and Ruixuan Li[1*]

[1]School of Computer Science and Technology, Huazhong University of Science and Technology,
[2]iWudao Tech
[1]{anders, yixiongz, xy_fu, rxli}@hust.edu.cn, weiliumg@gmail.com [2]jwang@iwudao.tech

## Abstract

Out-of-distribution generalization under distributional shifts remains a critical challenge for graph neural networks. Existing methods generally adopt the Invariant Risk Minimization (IRM) framework, requiring costly environment annotations or heuristically generated synthetic splits. To circumvent these limitations, in this work, we aim to develop an IRM-free method for capturing causal subgraphs. We first identify that causal subgraphs exhibit substantially smaller distributional variations than non-causal components across diverse environments, which we formalize as the Invariant Distribution Criterion and theoretically prove in this paper. Building on this criterion, we systematically uncover the quantitative relationship between distributional shift and representation norm for identifying the causal subgraph, and investigate its underlying mechanisms in depth. Finally, we propose an IRM-free method by introducing a norm-guided invariant distribution objective for causal subgraph discovery and prediction. Extensive experiments on two widely used benchmarks demonstrate that our method consistently outperforms state-of-the-art methods in graph generalization. Code is available at `https://github.com/anders1123/IDG`.

## 1 Introduction

Graph Neural Networks (GNNs) have demonstrated exceptional performance in various graph tasks [17, 42, 45]. However, they typically assume that training and testing data are independent and identically distributed (the i.i.d assumption), a condition that often fails in real-world applications [13, 8]. When the testing distribution diverges from the training distribution, model performance can deteriorate substantially. This has motivated the community to investigate out-of-distribution generalization in GNNs, aiming to make models robust in unseen environments.

To address out-of-distribution generalization in graphs, most methods are designed to extract the task-relevant causal subgraph/subfeatures [5, 4, 6, 9, 22, 32, 44, 47, 48]. They typically adopt the framework of Invariant Risk Minimization (IRM) [1, 18] to train a classifier that maintains predictive consistency across environments, thereby capturing causal features. Despite achieving promising results, IRM requires explicit environment information, such as carefully divided environment data and labels, which is costly to obtain. Alternative approaches for generating synthetic environments via prediction or perturbation also face inherent limitations. This raises a challenging research question: *can we circumvent IRM and capture the causal subgraph?*

---

*Corresponding authors. This paper is a collaboration between Intelligent and Distributed Computing Laboratory, Huazhong University of Science and Technology and iWudao Tech.

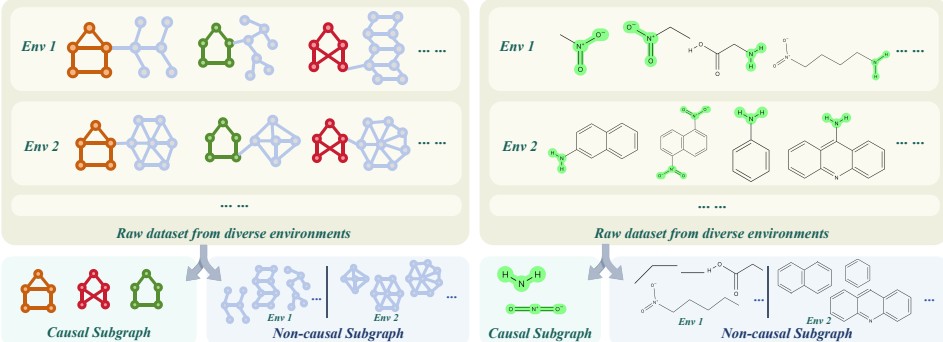

Figure 1: Our intuition: causal subgraphs exhibit smaller distributional shifts across environments than non-causal ones. For instance, in Motif, the causal subgraph comprises three typical motifs across environments, while the base graph shows diverse. Similarly, molecular properties rely on a small, stable set of substructures while non-causal components vary widely across environments.

Toward this end, in this work, we identify an intuitive phenomenon for detecting causal subgraphs without IRM requirements: the causal subgraph varies far less than the non-causal one across environments. As shown in Figure 1, in the synthetic Motif dataset [8], the causal subgraph always comprises the three canonical motif structures, while non-causal components vary across environments. Similarly, in chemical molecules, properties often rely on a limited set of substructures (termed fingerprints or moieties [38]) that remain stable across environments, whereas the remaining parts vary markedly. Based on this observation, we propose the following hypothesis: **causal subgraphs exhibit significantly smaller distributional shifts across environments than non-causal ones**, which uncovers an approach for extracting causal subgraphs while bypassing IRM. In this paper, we first establish it as the **Invariant Distribution Criterion** and offer a theoretical justification for it.

However, quantifying distributional shifts of subgraphs is scarcely addressed in graph learning. Drawing on prior work in other fields [16, 30, 23, 24, 25, 26], we demonstrate that, in graph models, distributional shifts lead to a reduction in the activations and representation norms of model outputs similarly. Furthermore, we find the quantitative relationship between distributional shift and samples' representation norms: activations and representation norms systematically decay as distribution shift intensifies, and we investigate the underlying mechanisms. This quantitative relationship indicates that representation norms can serve as a proxy for quantifying the extent of distributional shifts in input subgraphs.

Building on these insights, we propose a innovative objective for identifying causal subgraphs: the norm-guided invariant distribution objective, which maximizes subgraph representation norms to constrain and minimize subgraph cross-environment shift, and extract causal subgraphs efficiently without IRM. Leveraging this objective, we develop a novel framework for causal subgraph discovery and prediction. The main contributions are as follows:

• We introduce and theoretically validate the **Invariant Distribution Criterion**, revealing that causal subgraphs exhibit significantly smaller distributional shifts across environments than non-causal subgraphs. This criterion holds naturally under distributional shifts, without necessitating the explicit environmental requirements of IRM.

• We systematically uncover the quantitative relationship between distributional shift and representation norm in graph data for identifying causal subgraphs, and investigate its underlying mechanisms.

• Grounded in the criterion and quantitative relationship, we propose an innovative IRM-free method termed **IDG** (**I**nvariant **D**istribution **G**eneralization) by introducing a norm-guided invariant distribution objective for causal subgraph discovery and prediction.

• Extensive experiments on two widely used benchmarks demonstrate that our method surpasses the state of the art in graph out-of-distribution generalization.

## 2  A Closer Investigation into Invariant Distribution

This section is structured as follows: Section 2.1 presents the preliminaries; Sections 2.2 and 2.3 provide theoretical and technical insights of invariant distribution, respectively; and Section 2.4 introduces the final practical paradigm for out-of-distribution generalization.

## 2.1 Preliminaries

**Notations:** Let $G := (\mathcal{V}, \mathcal{E})$ be an undirected graph with $n$ nodes and $m$ edges, represented by its adjacency matrix $A$ and node feature matrix $X \in \mathbb{R}^{n \times d}$ with $d$ feature dimensions. We write $G_c$ and $G_s$ for the invariant (causal) and spurious subgraphs of $G$. $g_\theta : \mathcal{G} \mapsto \mathcal{G}$ and $h_\phi : \mathcal{G} \mapsto \mathcal{Y}$ denote the subgraph extractor and predictor with parameter $\theta, \phi$ respectively, and $Z$ represents the extracted subgraph $Z$ given by $Z = g_\theta(G)$. $\widehat{Y}_Z$ is the predicted label given by $\widehat{Y}_Z = h_\phi(Z)$ while $Y$ is the true label. We denote the training and testing set as $\mathcal{D}_{tr} = \{G^e\}_{e \in \mathcal{E}_{tr}}, \mathcal{D}_{te} = \{G^e\}_{e \in \mathcal{E}_{te}}, \mathcal{E}_{tr} \neq \mathcal{E}_{te}$.

**Problem Definition:** We focus on OOD generalization in graph classification. Given a collection of graph datasets $\mathcal{D} = \{G^e\}_{e \in \mathcal{E}_{tr} \subseteq \mathcal{E}_{all}}$, the objective of OOD generalization on graphs is to learn an optimal GNN model $f^*(\cdot) : \mathcal{G} \to \mathcal{Y}$ with data from training environments $\mathcal{D}_{tr} = \{G^e\}_{e \in \mathcal{E}_{tr}}$ that effectively generalizes across all (unseen) environments:

$$f^*(\cdot) = \arg\min_f \sup_{e \in \mathcal{E}_{all}} \mathcal{R}(f \mid e), \tag{1}$$

where $\mathcal{R}(f \mid e) = \mathbb{E}^e_{G,Y}\big[\ell\big(f(G), Y\big)\big]$ is the risk of predictor $f(\cdot)$ in environment $e$, and $\ell(\cdot, \cdot)$ denotes the loss function. Specifically, $f(\cdot) = g_\theta \circ h_\phi$ in subgraph-based methods.

**Invariant Risk Minimization** (IRM) [1] is a learning principle that seeks a feature representation on which a single classifier remains simultaneously optimal across multiple training environments, thereby capturing causal features and improving robustness to distribution shifts. In its ideal form, IRM solves the bi-level problem:

$$\min_{\phi, w} \sum_{e \in \mathcal{E}} R^e\big(w \circ \phi\big) \quad \text{s.t.} \quad w \in \arg\min_{\bar{w}} R^e\big(\bar{w} \circ \phi\big) \text{ for all } e \in \mathcal{E}, \tag{1}$$

where $\phi$ is a feature extractor, $w$ is a classifier, and $R^e$ the risk in environment $e$.

To enforce classifier optimality across different sub-distributions (the $argmin$ constrait), IRM partitions the training data into explicit environment-specific subsets, requiring explicit environment labels which is typically unavailable in practice. More details of IRM are included in Appendix B.

**Data Generating Process** in our work follows prior study [4, 9, 22, 32, 47, 48] and is founded on Structural Causal Models [34], which elucidate how latent factors give rise to observable graph properties. We model graph creation via a function $f_{\text{gen}} : \mathcal{Z} \to \mathcal{G}$. where $\mathcal{Z} \subseteq \mathbb{R}^n$ is the latent space and $\mathcal{G}$ the space of graphs. Under this SCM perspective, generation decomposes into three stages—first producing the causal subgraph $G_c$, then the spurious subgraph $G_s$, and finally the full observed graph $G$:

$$G_c := f^{G_c}_{\text{gen}}(C), \quad G_s := f^{G_s}_{\text{gen}}(S), \quad G := f^G_{\text{gen}}(G_c, G_s), \tag{2}$$

Let $C$ and $S$ be latent codes for causal and spurious factors. The observed graph G splits into a causal subgraph $G_c$ (from $C$) and a spurious subgraph $G_s$ (from $S$). $C$ drives the target $Y$, while $S$ vary across environments. Prior works distinguish Fully Informative Invariant Features(FIIF) when $Y \perp S|C$ and Partial Informative Invariant Features(PIIF) when $Y \not\perp S|C$ (refer to Appendix B).

## 2.2 Theoretical Insights: Invariant Distribution Criterion

As illustrated by the intuitive phenomenon in Figure 2, **causal subgraphs exhibit significantly smaller distributional shifts across environments than non-causal ones.** In this section, we present the theoretical proof of this hypothesis, termed **Invariant Distribution Criterion**.

We consider an input graph $G$ with associated label $Y$, observed under multiple environments $e \in \mathcal{E}$. Let $G_c$ denote the causal subgraph of $G$ that contains the true causal features determining $Y$, and let $G_s = G \setminus G_c$ be the remaining non-causal (spurious or confounded) parts of the graph. As in previous work of graph generalization, we here disregard the effects of noise and other irrelevant factors. We formalize the problem with the following assumptions:

**Assumption 1.** *Causal Mechanism: The label $Y$ is generated from $G$ exclusively via $G_c$. In particular, there exists a fixed causal mechanism $Y = f(G_c)$ that is invariant across environments. Equivalently, $Y \perp e \mid G_c$ for all environments $e$, meaning the conditional distribution $P_e(Y \mid G_c)$ is the same for every $e \in \mathcal{E}$. (In other words, $G_c$ contains all the information needed to determine $Y$, and this causal relationship does not change with the environment.)*

This assumption means that the generation of causal labels is fully determined by the causal subgraph $G_c$. This assumption derives from the Independent Causal Mechanism (ICM) hypothesis [35] and underpins Graph OOD methods such as CIGA [5] and LECI [9]. Invariant Risk Minimization (IRM) likewise relies on it by seeking representations that yield the same optimal classifier across environments. For example, if the causal subgraph (e.g., functional groups responsible for key chemical properties) is known, one can infer a molecule's chemical properties regardless of its distribution of origin (environments).

**Assumption 2.** *Environmental Diversity and Interventions: Across different environments, the distribution of the input graph $G$ may change due to interventions or context changes, but these changes are sparse in the sense of affecting only parts of the data-generating process at a time. In particular, we assume that environmental changes tend to affect the non-causal parts $G_s$ more significantly (or independently) than the causal part $G_c$.*

This assumption originates from Invariant Causal Prediction (ICP) framework and its extensions [34, 11, 36, 3], this assumption holds that distributional shifts across environments arise from sparse interventions on the non-causal subgraph $G_s$, while $G_c$ remains stable. This hypothesis has been adopted by Graph OOD methods such as LECI, which impose an even stronger assumption—that the environment $E$ is independent of $G_c$. i.e. $E \perp G_c$. For instance, the types and structures of functional groups in a molecule are limited and stable, whereas non-causal parts (e.g., heteroatoms) vary widely across environments.

**Assumption 3.** *Effective Classifier Support: We assume the classifier $h_\phi(\cdot)$ (to be learned) is powerful enough to model the true causal relationship $f(G_c)$. The classifier is trained on data from some source environment(s) and achieves low error on those. We further assume that if the test environment provides inputs $(G_c, G_s)$ whose causal subgraph component $G_c$ lies within the support of the training distribution of $G_c$, then the classifier can classify such inputs effectively. In contrast, if the test data's $G_c$ lies far outside the training support (an out-of-support scenario), no classifier can be expected to perform well without additional extrapolation assumptions.*

This assumption is based on the classical domain-adaptation bounds presented in [2], this assumption requires overlap between the support of source and target distributions to guarantee generalization. Since classification relies on $G_c$, the support of the causal-subgraph distribution in the test domain must lie within that of the training domain. For example, a classifier can correctly and consistently label a functional group in the test environment only if that group appeared during training.

Under the framework of the data generation process outlined in Section B, we provide theoretical guarantees to address the challenges associated with subgraph discovery. We initiate our analysis with these lemmas if the assumptions hold:

**Lemma 1.** *Invariant Conditional Distribution: Under Assumption 1, the conditional distribution of the label given the causal subgraph is invariant across environments: for all $e, e' \in \mathcal{E}$, $P_e(Y \mid G_c) = P_{e'}(Y \mid G_c)$. Equivalently, $Y$ depends on $G_c$ and not on $e$ or $G_s$. Moreover, no proper subset of features that excludes part of $G_c$ can enjoy this invariance, and any superset including non-causal parts $G_s$ will generally violate invariance.*

Lemma 1 follows directly from the SCM: under both PIIF and FIIF shifts, $C$ is the unique cause of $Y$. The formal proof of Lemma 1 is included in the Appendix C.1.

**Lemma 2.** *Domain Adaptation Bound for Representation Shift: For any representation $Z = \phi(G)$ used by a classifier $h_\phi(\cdot)$, let $R_e(h_\phi)$ denote the classification risk (error rate) in environment $e$. For any two environments $e$ (source) and $e'$ (target), the difference in risk is bounded by(take binary classification as an example):*

$$R_{e'}(h_\phi) \leq R_e(h_\phi) + \frac{1}{2} d_{\mathcal{H}\Delta\mathcal{H}}\big(P_e(Z),, P_{e'}(Z)\big) + \lambda^* \tag{3}$$

$d_{\mathcal{H}\Delta\mathcal{H}}$ is the $\mathcal{H}\Delta\mathcal{H}$ divergence (distribution discrepancy) between the source and target distributions, and $\lambda^* = \min_{h'}\big(R_e(h') + R_{e'}(h')\big)$ is the minimal combined error achievable on both domains (accounting for labeling function differences) . In particular, if the labeling function is invariant (as with $Z = G_c$ by Lemma 1) then $\lambda^* = 0$ (no intrinsic target error beyond distribution shift).

**Lemma 3.** *Support Overlap and Classifier Validity: If the support of the target environment's $Z$ distribution lies within (or significantly overlaps) the support of the source distribution, then any*

*classifier $h_\phi(\cdot)$ that is consistent on the source support can, in principle, maintain its performance on the target. Conversely, if the target $Z$ distribution produces samples outside the source support (out-of-support region), then no learning algorithm trained only on source data can guarantee accurate classification on those novel samples.*

Lemma 2 is drawn from the established theory of domain generalization [2]. Lemma 3 posits that by extracting the causal subgraph $G_c$, the model is guaranteed to operate within the support of familiar and stable features (avoiding out-of-support samples caused by novel non-causal feature combinations) and thus preserves generalization under distribution shift. The formal proof of Lemma 2 and 3 is included in the Appendix C.2 and C.3.

With these lemmas in hand, we now present the main theoretical statements about the invariant distribution criterion for causal subgraph:

**Theorem 1.** *Causal Subgraph Minimizes Distribution Shift Across Environments*

*For any two different environments $e, e'$, consider a measure $\Delta(G) := d(P_e(G), P_{e'}(G))$ of distribution shift for some divergence $d(\cdot, \cdot)$(e.g. $\mathcal{H}\Delta\mathcal{H}$ distance), for any alternative subgraph $G'$ that is not purely the causal subgraph, we have:*

$$\Delta(G_c) < \Delta(G') \tag{4}$$

*Equivalently, extracting $G_c$ leads to minimal distribution disparity between environments, whereas any inclusion of non-causal parts or exclusion of causal parts increases distribution shifts.*

**Theorem 2.** *Causal Subgraph Ensures Support Coverage and Stable Performance*

*If the extracted subgraph $Z$ is the real causal $G_c$, then in any new environment $e'$, its distribution will remain within the training support (or a reasonable interpolation range). Consequently, a classifier $h_\phi(\cdot)$ trained on $G_c$ in the source environment will retain its performance in $e'$, assuming the causal link remains unchanged. In contrast, using a non-causal subgraph $G'$ may produce out-of-support subgraph inputs in $e'$, i.e., out-of-distribution for $h_\phi(\cdot)$, leading to failures and unstable results.*

We defer the proofs and accompanying discussion of Theorems 1 and 2 to the Appendix C.4 and C.5.

Theorem 1 and 2 have formally shown that the causal subgraph $G_c$ of an input graph $G$ provides an invariant and sufficient representation for out-of-distribution generalization. Theorem 1 confirmed that $G_c$ experiences the least distribution shift across environments compared to any representation entangled with non-causal parts. Theorem 2 established that using $G_c$ ensures (causal subgraphs distribution) in new environments stay within the classifier's support, leading to stable performance. Theorem 1 and 2 provides a theoretically rigorous proof of the proposed Invariant Distribution Criterion, while guaranteeing the generalizability of the causal subgraph.

### 2.3 Technical Insights: Relation between Distributional Shift and Representation Norm

While the Invariant Distribution Criterion offers a pathway for circumventing IRM, it specifies a challenge for *quantifying* distributional shifts for graph learning. In this Section, we establish a pioneering quantitative connection between distributional shifts and representation norms in GNNs: **activations and representation norms systematically decay as distribution shift intensifies**.

#### 2.3.1 Norm and Activation Reduction when Distribution Shift Occurs

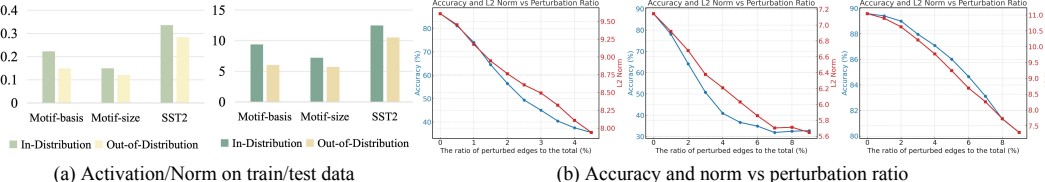

Figure 2: (a) Average activation values and L2 norms in GNN on training (in-distribution) and test sets (out-of-distribution). (b) Accuracy and norms under varying perturbation ratios—higher ratios indicate greater distributional shifts. Motif-basis, Motif-size, and Graph-SST2 from left to right.

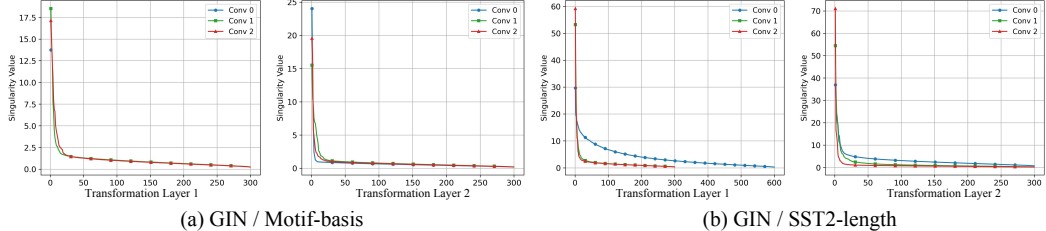

(a) GIN / Motif-basis

(b) GIN / SST2-length

Figure 3: SVD results of weight matrices in GIN (3 graph convolutional layers, each consists of 2 transforms) trained with ERM objective. Singular values are sorted in descending order.

Prior work in other research area has shown that distributional shift may manifest in network activations and representation norms [16], but the universality of this phenomenon has never been verified on graph learning.

We begin our investigation on the basic GNN. We train the GNN with the empirical risk minimization (ERM) objective on in-distribution data and evaluate it on OOD. As shown in Figure 2 (a), we find:

**Finding 1: Activations and representation norms diminish under distributional shift.**

Furthermore, to quantify how shift severity affects representation norms, we consider varying degrees of structural shift. Since manually curating datasets with precise shift levels is impractical, we adopt an alternative inspired by prior work: emulate structural shifts of differing severity by randomly delete and generate a fixed proportion of edges (not change edge count). From Figure 2 (b), we find:

**Finding 2: Increasing shift severity leads to progressively lower representation norms and a corresponding drop in predictive accuracy.**

Findings 1 and 2 show that the representation norm steadily decreases as the severity of input distribution shifts increases, suggesting that the norm can serve as a proxy for quantifying distributional shift. In the following Section, we further examine the underlying mechanism of this quantitative relation.

### 2.3.2 Why Distributional Shifts are Reflected in Norms

Prior studies indicate that the relationship between input distributions and output activations can be driven by the *low-rank* property of neural network weight matrices[16]. Low-rank refers to the case where a layer's weight matrix can be approximated by a matrix with lower rank, leading the network to concentrate on a limited set of directions. However, this property has not yet been confirmed for graph neural networks (GNNs). We find the same behavior in the weight matrices of GNNs, as shown in Figure 3. Singular value decomposition results reveals that only a small subset of singular values is large, while the remainder are relatively small or near zero, demonstrating that the network attends to a constrained set of directions. More results are included in the Appendix E.

Specifically, owing to the low-rank nature of neural networks—where weight matrices in certain layers can be approximated or inherently represented by lower-rank factors—the weights in networks attend only to a limited set of principal directions. When inputs within or near the training support is mapped by such a weight matrix, its energy (or norm) is preserved primarily along those directions that are amplified or effectively transmitted, yielding higher activation values. Under substantial distribution shift, however, the dominant components of shifted inputs may fail to project efficiently onto the low-dimensional subspace to which the network weights are "tuned" during training. In other words, these input features misalign with the weight matrix's principal directions, resulting in smaller projections and consequently lower activations—overall, the representation norm decreases. We present more example to illustrate how different inputs manifest in norm in Appendix F.

### 2.4 Practical Paradigm: Norm-Guided Invariant Distribution Objective

Based on the theoretical and quantitative insights in Section 2.2 and 2.3, we can draw the following inference: Causal features remain aligned while spurious or misaligned features collapse in distinct environments, yielding a higher representation norm for the causal components. Leveraging this insight, we introduce a novel paradigm for extracting causally invariant subgraph $Z$, termed Norm-Guided Invariant Distribution Objective. The objective is to maximize the representation norm of the extracted subgraph $Z$ while minimizing the distributional shift across environments, formulated as:

**Norm-Guided Invariant Distribution Objective:**

$$\max_{Z} \quad \Big\{ \underbrace{\mathbb{E}\big[Norm(H_Z)\big]}_{\text{Minimal Distribution Shift}}, \quad \underbrace{I(Z;Y)}_{\text{Stable Prediction}} \Big\} \tag{5}$$

$$\text{s.t.,} \ \ Z = g_\theta(G), \ (G, Y) \sim \mathcal{D}_{e_1, e_2}, e_1 \neq e_2$$

$Z$ is the extracted subgraph, $H_Z$ is the representation given by the preditcor $h_\phi$ and $I(;)$ is mutual information. Intuitively, the network's feature norm can be viewed as a proxy for how much signal from the input is influencing the prediction. The causal subgraph, being the truly predictive part of the input, is the only subset that can consistently supply strong signal across environments. Spurious features might look useful in a training set of a definite environment, but when the environment changes, the network effectively "turns down the volume" on those inputs – resulting in small-norm features and a prediction that defaults to a constant, since the distribution of spurious features is less stable than that of causal features, according to Theorem 1. The causal features, by contrast, continue to drive high-magnitude representations and confident predictions in different environments. Therefore, optimizing for maximal feature norm in the classifier naturally leads it to rely on the causal subgraph. This theoretical result aligns with the idea that causal features are the most stable and predictive presented in Theorem 2, while non-causal correlations get suppressed under shift, manifesting as low-norm, non-informative representations. Note that our method merely requires the dataset to include samples from multiple environments (i.e., to exhibit environment shift) and does not rely on explicitly partitioning environments as in IRM.

## 2.5 Conclusion

Based on the above theoretical results, experiments, and analyses, we draw the following conclusions: (1) Causal subgraphs exhibit significantly smaller distributional shifts across environments than non-causal ones (Section 2.2). (2) Representation norms, which systematically decay as distribution shift intensifies, can serve as a proxy for quantifying distributional shift for identifying causal subgraphs (Section 2.3). (3) Therefore, by maximizing subgraph representation norms to constrain and minimize cross-environment shift, causal subgraphs can be extracted efficiently. (Section 2.4).

## 3 Methodology

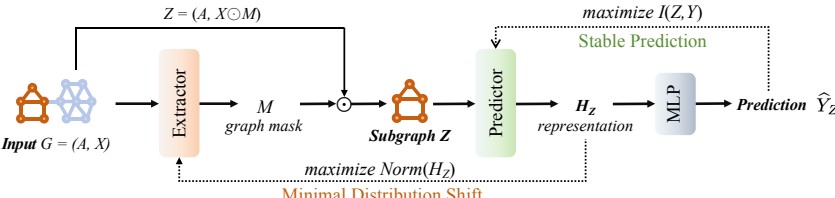

Figure 4: IDG framework. The extractor selects the subgraph $Z$ and feeds it to the predictor. Subgraph distributional shift are minimized by maximizing the representation norm of $Z$.

In this section, we propose a novel method, termed **IDG** (**I**nvariant **D**istribution **G**eneralization), for attaining the Norm-Guided Invariant Distribution Objective.

**Framework:** Based on the basic extract–predict framework as presented in Figure 4. Noting the imput graph as $G = (A, X)$, where $A$ is the adjacency matrix and $X$ represents node attributes. The subgraph extractor $g_\theta : \mathcal{G} \mapsto \mathcal{G}$ is tasked with selecting the optimal subgraph $Z$ from the input graph. We employ the Graph Isomorphism Network (GIN) as the backbone of the extractor and predictor. Specifically, the subgraph extractor generates a representation for each node:

$$H_G = g_\theta(G) \in \mathbb{R}^{n \times d'} \tag{6}$$

$d'$ is the representation feature dimension. For each edge $e = (i, j)$, the representation of node $i$ and node $j$ are concatenated and transformed to get the importance score of the edge:

$$M_{ij} = \sigma(\text{MLP}_1([H_i \| H_j])), \quad \forall e = (i, j) \in G \tag{7}$$

$$Z = (A_c, X),^1 \quad where \quad A_c = \text{Top}_r(M \odot A) \tag{8}$$

Here $\text{Top}_r(\cdot)$ selects the top $r$ edges with the highest importance scores, adopted from in previous work [22]. $r$ is a predefined hyperparameter commonly set to 0.5. The extracted subgraph $Z$ is then fed into the predictor $h_\phi$ to make representations and final predictions:

$$H_Z = \text{READOUT}(h_\phi(Z)) \in \mathbb{R}^{d'}, \quad \widehat{Y}_Z = \text{MLP}_2(H_h) \tag{9}$$

**Optimization:** Guided by the Norm-Guided Invariant Distribution Objective in Equation 5, the subgraph extractor identifies the causal subgraph that minimizes distributional shift (i.e., maximizes the representation norm), while the predictor infers the corresponding labels. We then reformulate the optimization objective of Equation 5 into the following practical scheme:

$$\textbf{Extractor:} \quad \mathcal{L}_\theta = CE(\widehat{Y}_Z, Y) + \lambda_1 \cdot [-log(\|H_Z\|_2)] + \lambda_2 \cdot \mathcal{L}_{comp}$$
$$\textbf{Predictor:} \quad \mathcal{L}_\phi = CE(\widehat{Y}_Z, Y) \qquad \textbf{s.t.,} \ Z = g_\theta(G), \quad (G, Y) \sim \mathcal{D}_{e_1, e_2}, \tag{10}$$

The coefficient $\lambda_1, \lambda_2$ is a task-dependent balancing parameter. $CE(\cdot, \cdot)$ is the cross-entropy loss.

The $-log(\|H_Z\|_2)$ term, by maximizing the representation norm of the selected subgraph, compels the extractor to identify the subgraph exhibiting minimal distributional shift (i.e., the causal subgraph) in accordance with our **Invariant Distribution Criterion**, thereby resulting in stable predictions.

We also adopt the entropy term from prior work [50, 31] to encourage compactness of the extracted subgraphs: $\mathcal{L}_{comp} = -[(1 - M) \log(1 - M) + M \log M]$

During training, since the extractor and the predictor optimize different objectives, we split each epoch into two stages: the extractor's parameters are first frozen while the predictor is updated via Equation 10; then the predictor's parameters are frozen and the extractor is updated. During inference, the subgraph extractor directly extracts the subgraph, and the predictor makes predictions based on it.

## 4 Experiments

### 4.1 Datasets, Metrics and Baselines

We adopt two widely used benchmarks for graph OOD generalization—GraphOOD [8] and DrugOOD [15], across seven datasets: Motif, CMNIST, HIV, SST2, and Twitter from GraphOOD, and EC50 and IC50 from DrugOOD. These datasets span synthetic, superpixel, molecular, and text graphs. Each dataset contains one or more domains and is divided into domain-based splits, thereby introducing distribution shifts. ROC-AUC metric is used for the binary classification dataset and Accuracy for the others. Refer to the Appendix H for dataset details.

Following [9], we employ GIN for both the extractor and predictor, set $(\lambda_1, \lambda_2) = (0.1, 0.01)$, and retain the original learning-rate and batch-size settings. We compare our method with several competitive baselines, including empirical risk minimization (ERM), four traditional out-of-distribution (OOD) baselines including IRM [1], VREx [19], and Coral [39], and eight graph-specific OOD baselines including DIR [44], GIL [22], GSAT [32], CIGA [5], LECI [9], iMoLD [51], EQuAD [47] and LIRS [49]. Refer to Appendix I for details about baselines.

### 4.2 Comparison with State-of-the-Art methods

Table 1 present a comparison with IDG and other OOD methods on the GraphOOD and DrugOOD datasets. From the results, we can conclude that IDG attains state-of-the-art performance on 15 out of 17 datasets and achieves comparable results on the other two, demonstrating its superior generalization ability across different types of datasets and domain shifts. Moreover, some OOD methods even underperform ERM on certain datasets because the complexity of the domain information precludes their simple division into a few explict environment splits, further underscoring the advantage of IDG.

### 4.3 Contribution of Norm-Guided Invariant Distribution Objective

To rigorously assess the impact of the proposed Norm-Guided Invariant Distribution Objective (hereafter "our objective"), we introduce a baseline model obtained by removing the norm term from

---

[1]We do not impose any restrictions on the node attributes $X$ because nodes with all edges masked will be excluded from the GNN's message-passing process, thus not influencing the results.

Table 1: Results on GraphOOD and DrugOOD dataset in 3 rounds.

| Dataset | Motif | | CMNIST | HIV | | SST2 | Twitter | IC50 | | | EC50 | | |
|---|---|---|---|---|---|---|---|---|---|---|---|---|---|
| Domain | size | basis | color | scaffold | size | length | length | scaffold | size | assay | scaffold | size | assay |
| ERM | 53.46(4.08) | 63.8(10.36) | 27.82(3.24) | 69.55(2.39) | 59.19(2.29) | 80.52(1.13) | 57.04(1.70) | 68.79(0.47) | 67.50(0.38) | 71.63(0.76) | 64.98(1.29) | 65.10(0.38) | 67.39(2.90) |
| IRM | 53.68(4.11) | 59.93(11.46) | 29.04(2.10) | 70.17(2.78) | 59.94(1.59) | 80.75(1.17) | 57.72(1.03) | 67.22(0.62) | 61.58(0.58) | 71.15(0.57) | 63.86(1.36) | 59.19(0.83) | 67.77(2.71) |
| Coral | 53.71(2.75) | 66.23(9.01) | 29.47(3.15) | 70.69(2.25) | 59.39(2.90) | 78.94(1.22) | 56.14(1.76) | 68.36(0.61) | 64.53(0.32) | 71.28(0.91) | 64.83(1.64) | 58.47(0.43) | 72.08(2.80) |
| VREx | 54.47(3.42) | 66.53(4.04) | 27.65(2.31) | 69.34(3.54) | 58.49(2.28) | 80.20(1.39) | 56.37(0.76) | 67.32(0.53) | 63.47(0.41) | 70.53(0.86) | 63.63(0.96) | 59.89(0.41) | 69.28(2.34) |
| DIR | 44.83(4.00) | 39.99(5.50) | 26.20(4.48) | 68.44(2.51) | 57.67(3.75) | 81.55(1.06) | 56.81(0.91) | 66.33(0.65) | 62.92(1.89) | 69.84(1.41) | 63.76(3.22) | 61.56(4.23) | 65.81(2.93) |
| GIL | 53.92(3.88) | 64.23(5.98) | 27.13(2.17) | 69.43(2.31) | 59.27(3.39) | 80.43(1.73) | 55.40(2.64) | 65.38(0.72) | 63.06(1.92) | 69.71(1.63) | 62.56(3.84) | 61.73(3.36) | 66.84(2.27) |
| GSAT | 60.76(5.94) | 55.13(5.41) | 35.62(5.52) | 70.07(1.76) | 60.73(2.39) | 81.49(0.76) | 56.07(0.53) | 66.45(0.50) | 66.70(0.37) | 70.59(0.43) | 64.25(0.63) | 62.65(1.79) | 73.82(2.62) |
| CIGA | 54.42(3.11) | 67.15(8.19) | 32.11(2.53) | 69.40(1.97) | 59.55(2.56) | 80.46(2.00) | 57.19(1.15) | 69.14(0.70) | 66.92(0.54) | 71.86(1.37) | 67.32(1.35) | 65.65(0.82) | 69.15(5.79) |
| LECI | 71.43(1.96) | 73.16(2.22) | 51.80(2.53) | 71.36(1.52) | 65.44(1.78) | 83.44(0.27) | 57.63(0.14) | / | / | / | / | / | / |
| iMoLD | 58.23(0.43) | 65.58(1.27) | 48.35(2.44) | 72.93(2.29) | 62.86(2.58) | 82.13(0.69) | 56.46(1.74) | 68.84(0.58) | 67.92(0.43) | 72.11(0.51) | 67.79(0.88) | 67.09(0.91) | 77.48(1.70) |
| EQuAD | 59.72(3.69) | 67.11(10.11) | 48.98(2.36) | 72.24(0.64) | 64.19(0.56) | 82.57(0.36) | 57.47(1.43) | 69.27(0.86) | 68.19(0.24) | 73.26(0.47) | 68.12(0.48) | 66.37(0.64) | 79.36(0.73) |
| LIRS | 74.95(7.69) | 75.51(2.19) | 49.87(2.62) | 72.82(1.61) | 66.64(1.44) | 82.48(0.79) | 58.29(1.03) | 69.78(0.41) | 68.32(0.33) | 72.56(0.83) | 68.17(0.46) | 67.23(0.54) | 79.46(1.58) |
| IDG | 73.23(3.21) | 82.53(3.28) | 55.32(3.67) | 73.24(0.68) | 67.44(2.32) | 83.67(0.32) | 59.76(0.83) | 69.97(0.31) | 69.02(0.23) | 72.86(0.54) | 68.32(0.46) | 68.03(0.31) | 80.54(0.67) |

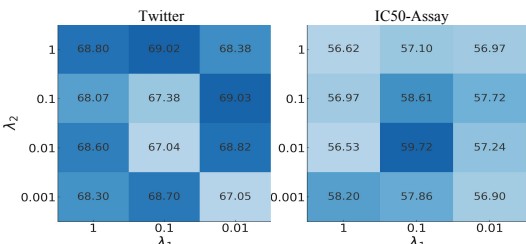

Figure 5: From left to right: (1) Training loss vs. epochs in IDG and baseline (2) Testing loss vs. epochs in IDG and baseline (3) Testing Accuracy vs. epochs in IDG and baseline (4) t-SNE results of IDG representations (5) t-SNE results of baseline representations

Equation 10, and both models were trained and evaluated on the ground-truth-annotated Motif-basis dataset. From the results in Figure 5 and Table 2, we draw the following conclusions:

**Our objective accurately captures distributional shifts.** Although the IDG and baseline model exhibit nearly identical loss trajectories on the training set, the testing loss in baseline anomalously increases in the test set (out-of-distribution), whereas the IDG's testing loss continues to decline in tandem with its training (Figure 5 (1-2)). These results indicate that our objective captures distributional shifts more accurately.

**Our objective significantly enhances out-of-distribution generalization.** In Figure 5(3), adding our objective accelerates convergence and delivers improved performance. Moreover, t-SNE visualizations (Figure 5 (4-5)) reveal that IDG produces more distinct embeddings for out-of-distribution samples than the baseline.

**Enhanced causal subgraph extraction.** As shown in Table 2, subgraphs extracted by IDG align more accurately with (edge) ground-truth, demonstrating that Our Objective steers the model to capture causal subgraph faithfully.

### 4.4 Extended Verification

**Hyperparameter Sensitivity** To assess IDG's sensitivity to hyperparameters, we conducted experiments on Twitter and IC50-assay. The results in Figure 6 (a) demonstrate that IDG is insensitive to the choice of parameters, indicating its stability across different configurations.

**Ablation Study** We conduct an ablation study

Table 2: Results with ground-truth on Motif.

| | Motif-basis | | | | Motif-size | | | |
|---|---|---|---|---|---|---|---|---|
| | acc | recall | pre | f1 | acc | recall | pre | f1 |
| Baseline | 0.6985 | 0.4004 | 0.6907 | 0.5068 | 0.9064 | 0.2609 | 0.3164 | 0.2860 |
| **IDG** | **0.7337** | **0.4444** | **0.7381** | **0.5547** | **0.9373** | **0.2967** | **0.3408** | **0.3171** |

Table 3: Ablation study results.

| Method | Motif-basis | CMNIST | Twitter | EC50-assay |
|---|---|---|---|---|
| ERM | 53.46 | 27.82 | 57.04 | 67.39 |
| ERM+$\mathcal{L}_{Norm}$ | 53.25 | 27.34 | 57.69 | 67.48 |
| w/o $\mathcal{L}_{CE}$ | 66.58 | 48.32 | 57.47 | 69.02 |
| w/o $\mathcal{L}_{Comp}$ | 75.41 | 52.53 | 58.89 | 78.96 |
| w/o $\mathcal{L}_{Norm}$ | 76.12 | 49.64 | 58.43 | 76.64 |
| IDG | **77.83** | **55.32** | **59.64** | **80.54** |

Figure 6: Hyperparameter sensitivity on $\lambda_1, \lambda_2$.

to evaluate the impact of different components of IDG. The results are shown in Table 3. In the table, ERM and ERM+$\mathcal{L}_{Norm}$ correspond to a GIN trained with ERM and ERM augmented by the invariant distribution objective, respectively. The results show that merely increasing the representation norm hardly improves generalization. w/o $\mathcal{L}_{CE}$, w/o $\mathcal{L}_{Comp}$ and w/o $\mathcal{L}_{Norm}$ denote training without the cross entropy, compactness constraint and norm for the extractor respectively. The results indicate that all objectives enhance performance, and confirm the effectiveness of our objective.

**Efficiency Study** The time complexity of the IDG method is $O(md+nd^2)$, where $n$, $m$, and $d$ denote the average number of nodes, edges, and feature dimensions per graph. Specifically: (1) message passing and node updates per layer incur a cost of $O(md+nd^2)$. (2) edge scoring and top-r selection can be completed in $O(m \log m)$ in the worst case. (3) norm regularization and

Table 4: Training and inference time (ms).

| Method | Motif-basis | | HIV-scaffold | | SST2-length | |
|--------|-------|-----------|-------|-----------|-------|-----------|
| | Train | Inference | Train | Inference | Train | Inference |
| GIL | 56801 | 2037 | 102594 | 22057 | 86658 | 16472 |
| DIR | 14415 | 794 | 38477 | 1098 | 32392 | 9377 |
| **IDG** | **11262** | **493** | **35232** | **995** | **26326** | **7659** |

compactness term require $O(m)$ time. In our method, the norm calculation incurs minimal computational overhead. To further demonstrate its efficiency, we measured training time and inference time. The results are presented in Table 4. Extended results about visualizations, hyperparameter $r$ and efficiency are included in the Appendix J, K and L.

# 5 Related Work

**Out-of-distribution Generalization in Graph.** OOD generalization is a critical challenge in graph learning, where models trained on a specific data distribution often fail to generalize well to unseen distributions. IRM [1], which seeks to learn causally relevant representations that remain stable across different environments, is widely adopted in graph generalization [46, 51, 22, 21, 44, 29, 5]. These methods either depend on provided or predicted environment labels or on environment-specific causal assumptions, which limits their practical applicability. Although some methods [49, 47, 32, 43] extract causal features using Infomax or the Information Bottleneck principles without necessitating environment labels, they only partially address the causal–spurious distinction. To bridge this gap, we introduce an novel criterion via an entirely distinct paradigm. More discussions are in Appendix G.

**Low-Rank and Distribution Shift.** Previous studies have shown that trained deep neural networks generally exhibit low-rank property [14, 12, 41], but this phenomenon has not been rigorously evaluated in graph models. [16] demonstrated that distribution shifts manifest in the network's activations, which they attributed to the low-rank structure of neural weight matrices. However, these phenomena remain unexplored and unverified within graph model, and we provides a thorough analysis of this phenomenon in this work.

**Subgraph-based Methodology** Subgraph-based methods have gained prominence in graph out-of-generalization and explanation due to their ability to capture local structures and patterns. The extractor architecture employed in this work, which maps node features to edge masks, is also widely used in Graph OOD research such as DIR [44], GIL [22], LECI [9], CIGA [5], and GSAT [32] . Its origins trace back to earlier graph explanation methods like PGExplainer[50, 31, 37], as well as similar techniques in the text [20] and vision domains [37]. The central challenge of these methods is to establish a principled rigorous framework for accurately identifying causal sub-features, such as IRM and its variants adopted in prior works like DIR and GIL. In contrast, our method is completely IRM-free and requires no environment information, it identifies causal subgraphs simply by finding solutions that satisfy the minimal shift criterion via norm optimization.

# 6 Conclusion

In this paper, we propose Invariant Distribution Criterion, which demonstrate causal subgraphs undergo markedly smaller distribution shifts than non-causal ones. By linking representation norms to distribution shift, we derive a practical norm-based objective and instantiate it as IDG. Empirical results on diverse benchmarks show IDG consistently outperforms state-of-the-art OOD baselines.

# 7 Acknowledgement

This work is supported by the National Key Research and Development Program of China under grant 2024YFC3307900; the National Natural Science Foundation of China under grants 62436003, 62206102, 62376103 and 62302184; Major Science and Technology Project of Hubei Province under grant 2025BAB011, 2024BAA008; Hubei Science and Technology Talent Service Project under grant 2024DJC078. The computation is completed in the HPC Platform of Huazhong University of Science and Technology.

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

# A    Invariant Risk Minimization and Graph Out-of-Distribution Generalization

IRM [1] Invariant Risk Minimization (IRM) is a framework for learning predictors that remain robust under distribution shifts by enforcing that the same classifier is simultaneously optimal across multiple training environments. In its ideal form, IRM solves the bi-level problem

$$\min_{\phi,\, w} \sum_{e \in \mathcal{E}} R^e\big(w \circ \phi\big) \quad \text{s.t.} \quad w \in \arg\min_{\bar{w}} R^e\big(\bar{w} \circ \phi\big) \text{ for all } e \in \mathcal{E}, \tag{11}$$

where $\phi$ is a feature extractor, $w$ a classifier, and $R^e$ the risk in environment $e$. Many approaches for out-of-distribution generalization on graphs are based on the IRM framework. In practice, IRMv1 approximates this constraint by adding a penalty on the squared norm of the gradient of each environment's risk with respect to w, encouraging the learned representation to capture only invariant (i.e., causal) features:

$$\min_{\phi,w} \sum_{e \in \mathcal{E}} R^e\big(w \circ \phi\big) \; + \; \lambda \sum_{e \in \mathcal{E}} \left\| \nabla_{w|w=1} R^e\big(w \circ \phi\big) \right\|_2^2, \tag{12}$$

Graph out-of-distribution methods typically build on the frameworks of IRM, which seek to extract the causal subgraph/features and learn an equipredictive classifier across environments in order to capture invariant features, which demands that the dataset be divided into well-defined environments. Depending on the environmental partitioning strategy, these environments fall into three main categories: Approaches such as LECI [9] and G-splice [27] depend on environment labels provided in the dataset, but these labels are not always available and incur high annotation costs. Other methods such as GIL [22] and OOD-GCL [21] use unsupervised clustering to infer environment labels, which may not always align well with real environment distribution. Other approaches such as DIR [44] explicitly create distinct environments by applying causal interventions to the dataset. However, designing causal interventions to generate training distributions should require domain expertise or incur additional overhead for different task, and unreasonable designed interventions may fail to remove all spurious features or even damage crucial information[33]. Such limitations relying on environment information hinder the deployment of these methods in real-world scenarios. Recent work has further shown that recovering real environment information is infeasible without external information[28]. In summary, the limitations of these invariant learning methods have prompted us to explore an alternative approach for uncovering causal subgraphs.

# B    Graph Data Generation Process

In graph data generation process presented in 2, $C$ and $S$ denote latent codes for the causal and spurious factors, respectively. The observed graph $G$ is composed of two latent components: an causal subgraph $G_c$ driven by the causal factor $C$, and a spurious subgraph $G_s$ driven by the non-causal factor $S$, regardless of noise. The variable $C$ causally influences the target $Y$, whereas $S$ may vary across environments $E$. Depending on how $S$ interacts with $Y$ conditional on $C$, prior work typically distinguish two scenarios, i.e., (i) Fully Informative Invariant Features (FIIF) when $Y \perp S|C$ and (ii) Partial Informative Invariant Features (PIIF) when $Y \not\perp S|C$.

In case (i), the invariant factor $C$ is fully informative (FIIF) to the target label $Y$, and the latent spurious factor $S$ provide no further information. In case (ii), the invariant factor $C$ is only partially informative (PIIF) about $Y$, spurious factor $S$ can further provide additional information to aid the prediction of $Y$, however, as $S$ is directly affected by $E$, it is not stable across different environments. The SCMs for the two scenarios are illustrated in Figure 7, and these two assumptions have been extensively discussed and empirically validated in prior work on out-of-distribution graph tasks [4, 5, 6, 9, 22, 32, 44, 47, 48] and is founded on Structural Causal Models [34].

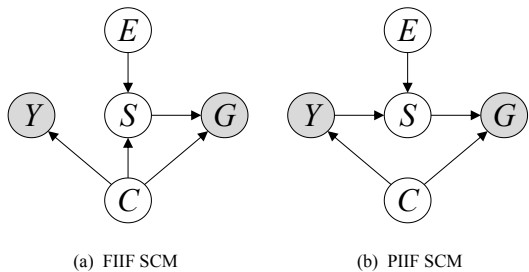

(a) FIIF SCM        (b) PIIF SCM

Figure 7: Structure causal models for graph data generation.

## C  Proofs for Theoretical Results

### C.1  Proof of Lemma 1

*Proof.* By Assumption 1, $Y = f(G_c)$ for a fixed causal mechanism $f(\cdot)$ that does not vary with e. This means that for any value $g_c$ of $G_c$, $P(Y \mid G_c = g_c)$ is defined entirely by $f(g_c)$ and is the same in every environment. More formally, for any measurable subset $A$ of the range of $Y$ and for any $g_c$, $P_e(Y \in A \mid G_c = g_c) = \mathbf{1}f(g_c) \in A$, which is evidently independent of $e$. Hence $P_e(Y \mid G_c) = P(Y \mid G_c)$ for all $e$. This captures the essence of invariant causal prediction, wherein the correct causal features $G_c$ yield a predictor that holds across domains .

Now, if we remove any component of $G_c$, the remaining features would be an incomplete causal subgraph, insufficient to fully determine $Y$. In that case, the conditional $P_e(Y \mid \tilde{G}_c)$ (with $\tilde{G}c \subsetneq G_c$) would generally depend on $e$ because the relationship between $\tilde{G}c$ and $Y$ could be confounded by the part of $G_c$ that is missing. Similarly, if we include any non-causal features from $G_s$ to form an augmented subgraph $G_c \cup G_s$, then $P_e(Y \mid G_c \cup G_s)$ may vary with $e$ because $G_s$ can carry spurious correlations with $Y$ that differ by environment. By Assumption 2, the correlation between $G_s$ and $Y$ is not stable: there exists at least two environments $e, e'$ for which $P_e(Y \mid G_s) \neq P_{e'}(Y \mid G_s)$ (since $G_s$ has no direct causal link to $Y$, any association is incidental and can change). Therefore, $P_e(Y \mid G_c, G_s)$ would generally differ from $P_{e'}(Y \mid G_c, G_s)$ because conditioning on $G_s$ can introduce environment-specific information. We conclude that only the true causal subgraph $G_c$ (or any superset that does not include spurious features) yields an invariant conditional for $Y$. $\square$

### C.2  Proof of Lemma 2

*Proof.* This is a standard result from domain adaptation theory [2]. We treat each environment as a domain with distribution $P_e(Z, Y)$. The $\mathcal{H}\Delta\mathcal{H}$-divergence between $P_e(Z)$ and $P_{e'}(Z)$ measures how well a classifier can distinguish between source and target representations; it can be seen as twice the supremum difference in probabilities assigned to sets by the two distributions (related to total variation distance restricted to hypothesis class $\mathcal{H}$). The cited bound (with $d(D_e, D_{e'})$ denoting this divergence) shows how much the source error can fail to transfer to target. For completeness: one derivation is

$$R_{e'}(h) - R_e(h) \leq \frac{1}{2}d_{\mathcal{H}\Delta\mathcal{H}}(P_e(Z), P_{e'}(Z)) + \lambda^*, \tag{13}$$

and similarly $R_e(h) - R_{e'}(h)$ is bounded by the same quantity, yielding the two-sided inequality mentioned in different form . The term $\lambda^*$ represents the best possible joint error; if the labeling rule is identical across domains, there exists a hypothesis (namely the Bayes-optimal classifier on that rule) that achieves low error on both, so $\lambda^*$ would be small (zero in the ideal case where Bayes error is zero for that representation). Under our Assumption 1, the same causal labeling function $f(G_c)$ applies in all environments, so for $Z = G_c$ one can achieve $\lambda^* = 0$ by choosing $h = f$. Thus, for the causal subgraph representation,

$$R_{e'}(h^*) \le R_e(h^*) + \frac{1}{2} d_{\mathcal{H}\Delta\mathcal{H}}(P_e(G_c), P_{e'}(G_c)), \tag{14}$$

with $h^*$ being the invariant optimal classifier. This quantifies that any degradation in accuracy is due solely to the shift in $G_c$'s distribution across $e$ and $e'$. This formal result confirms: an invariant representation (causal $G_c$) minimizes the transferable risk penalty to just the distribution divergence term, whereas a spurious representation incurs an additional irreducible error jump term. $\square$

### C.3 Proof of Lemma 3

*Proof.* This statement reflects a basic requirement for generalization under distribution shift. If $\mathrm{Supp}(P_{e'}(Z)) \subseteq \mathrm{Supp}(P_e(Z))$, the classifier $h_\phi(\cdot)$ trained on $P_e$ is at least receiving familiar inputs under $P_{e'}$. In the ideal case, if $h_\phi(\cdot)$ has learned the correct decision rule on $P_e(Z)$ (e.g. the true $f(\cdot)$ for $G_c$) and the rule remains the same (invariant labeling), it will apply equally to $P_{e'}(Z)$ as long as those inputs are not qualitatively new. Formally, for any $z$ in the target support, since $z$ is also in source support, $h_\phi(\cdot)$ had the opportunity to adjust its decision (or an equivalent $z$) during training; thus $(\cdot)$'s prediction at $z$ can be expected to be as reliable as in training. If the supports overlap heavily but not completely, we can expect performance to degrade gracefully in proportion to how much probability mass falls in the unfamiliar regions.

On the other hand, if $\mathrm{Supp}(P_{e'}(Z))$ extends to regions where $P_e(Z)$ has zero (or very low) density, then those $z$ values are effectively never seen during training. A classifier cannot be expected to extrapolate correctly to arbitrarily novel inputs without additional knowledge; in the worst case, an adversarially chosen out-of-support input could be assigned an incorrect label by $(\cdot)$ since $(\cdot)$ has no basis to learn the correct behavior there. In domain adaptation terms, when support does not overlap, the $\mathcal{H}\Delta\mathcal{H}$-divergence reaches its maximum (because a hypothesis can perfectly separate source and target supports), yielding a trivial bound $R_{e'}(h) \le R_e(h) + 1/2 \cdot 2 + \lambda^* = R_e(h) + 1 + \lambda^*$, which means essentially no guarantee of generalization. In summary, overlapping support is a necessary condition for successful transfer; without it, the new domain may contain feature patterns fundamentally outside the model's experience, leading to unpredictable performance. $\square$

### C.4 Proof of Theorem 1

*Proof.* We fix two arbitrary environments $e$ (source) and $e'$ (target) and compare the cross-environment divergence $\Delta(Z)$ for different choices of the subgraph $Z$. There are three typical cases to consider for an alternative subgraph $G'$ that is not equal to $G_c$:

**Case 1: $G'$ includes non-causal parts ($G_s$).** In this case, $G'$ can be viewed as $G' = G_c \cup U$ where $U \subseteq G_s$ is some subset of spurious features (or possibly all of $G_s$, including the trivial case $G' = G$). Because $G_s$ by definition contains the features that are not causally relevant to $Y$, any correlation between $U$ and $Y$ is spurious or environment-specific. By Assumption 2, environmental changes affect $G_s$ significantly; thus the marginal distribution of $U$ (and its correlation with $G_c$ or $Y$) varies across environments. This implies that the joint distribution $P_e(G_c, U)$ differs from $P_{e'}(G_c, U)$ to a greater extent than $P_e(G_c)$ differs from $P_{e'}(G_c)$. Intuitively, since $G_c$ is relatively stable but $U$ is highly variable across $e$ and $e'$, including $U$ will amplify the cross-environment disparity. Formally, most divergence measures are monotonic under the introduction of additional differing variables; for example, if $P_e(G_c) = P_{e'}(G_c)$ but $P_e(U) \ne P_{e'}(U)$, then the joint divergence satisfies $d\big(P_e(G_c, U), P_{e'}(G_c, U)\big) \ge d\big(P_e(U), P_{e'}(U)\big) > 0$. Even if $P_e(G_c)$ changes slightly across environments, the changes in $U$ (spurious part) are strictly larger (by Assumption 2), so $\Delta(G_c \cup U)$ will still exceed $\Delta(G_c)$.

In particular, consider the $\mathcal{H}\Delta\mathcal{H}$-divergence as the measure $d$. If $Z = G_c \cup U$ contains environment-varying spurious components, one can construct a hypothesis in $\mathcal{H}\Delta\mathcal{H}$ distance that focuses on $U$ to effectively distinguish which environment a sample came from. For instance, a classifier $h \in \mathcal{H}$ that predicts the environment identity from $U$ will achieve better-than-chance accuracy due to $U$'s distribution shift, implying a large $d_{\mathcal{H}\Delta\mathcal{H}}(P_e(Z), P_{e'}(Z))$. In contrast, if $Z = G_c$ (with all $G_s$ removed), then no classifier can reliably distinguish $e$ vs $e'$ because $G_c$ by itself varies minimally – in the ideal case, $P_e(G_c) = P_{e'}(G_c)$ if the causal features are entirely invariant. Thus $d_{\mathcal{H}\Delta\mathcal{H}}(P_e(G_c), P_{e'}(G_c))$ will be small (in fact zero if $G_c$'s distribution is truly identical across $e, e'$).

This reasoning formalizes that

$$\Delta(G_c \cup U) = d\big(P_e(G_c, U), P_{e'}(G_c, U)\big) > d\big(P_e(G_c), P_{e'}(G_c)\big) = \Delta(G_c).$$

Hence any inclusion of non-causal features $U$ increases the divergence across environments.

**Case 2: $G'$ excludes part of the causal subgraph.** In this scenario, $G'$ is a strict subset of $G_c$ (or possibly disjoint, but a disjoint subgraph would be pure $G_s$ which is covered by Case 1). Let $G_c = G' \cup C_{\mathrm{miss}}$, where $C_{\mathrm{miss}}$ is the portion of the true cause that is left out of $G'$. Because $G'$ is missing some of the true causal features, it no longer fully determines the label $Y$. In fact, by Lemma 1, $Y$ is not conditionally independent of the environment given $G'$ – since $G'$ omits part of $G_c$, the remaining features alone cannot guarantee the invariant relationship with $Y$. Equivalently, the effective labeling function on $G'$ (i.e. the relationship between $G'$ and $Y$) varies with the environment. In one environment, $Y$ may depend on $G'$ in one way, whereas in another environment the relationship shifts due to the influence of the missing causal factors $C_{\mathrm{miss}}$. This means there is no single classifier on $G'$ that perfectly fits $P(Y|G')$ across both $e$ and $e'$ – some environment-specific discrepancy in prediction is unavoidable.

Even if the marginal distributions of $G'$ happen to be similar across environments (for instance, if $P_e(G_c)$ itself is invariant or if the environment does not directly alter the observed part $G'$), the fact that $Y|G'$ differs implies a significant distribution shift in the joint distribution of features and labels. To see this, consider the joint divergence (e.g. total variation or KL) between $P_e(G', Y)$ and $P_{e'}(G', Y)$. We can decompose it as differences in the conditional label distributions: if there exists any $z'$ in the support of $G'$ for which $P_e(Y|G' = z') \neq P_{e'}(Y|G' = z')$, the joint distributions will differ. In quantitative terms, one can lower-bound, for example, the total variation distance by the average conditional difference:

$$\mathrm{TV}\big(P_e(G', Y), P_{e'}(G', Y)\big) \geq \frac{1}{2} \int_{z'} \big| P_e\big(Y \mid G' = z'\big) - P_{e'}\big(Y \mid G' = z'\big) \big| P_e(dz'). \quad (15)$$

$\mathrm{TV}(P, Q)$ is the total-variation distance between two probability measures $P$ and $Q$. By Lemma 1, such a difference is nonzero for $G'$ that excludes part of $G_c$ (there is at least some $z'$ for which the label distributions diverge across environments). Therefore, the joint distribution shift is positive. In contrast, for $Z = G_c$, we have $P_e(Y|G_c) = P_{e'}(Y|G_c)$ exactly (labeling function is invariant), so no such difference occurs and the joint distributions $P_e(G_c, Y)$ and $P_{e'}(G_c, Y)$ align on the conditional label component (any remaining shift comes only from $P(G_c)$ differences, which are small by Assumption 2).

From a domain adaptation viewpoint, the omitted causal features lead to an intrinsic labeling mismatch across domains. In the bound of Lemma 2, this manifests as a nonzero $\lambda^*$ term for $Z = G'$. In fact, $\lambda^*$ in inequality 3 represents the minimum combined error on both environments; if no single classifier can simultaneously achieve low error on both $e$ and $e'$ because the label mappings differ, then $\lambda^*$ is bounded away from 0. This contributes to an effective increase in distribution shift beyond what the feature divergence alone ($d_{\mathcal{H}\Delta\mathcal{H}}$) captures. Meanwhile, for $Z = G_c$, Lemma 1 guarantees the labeling function is identical in $e$ and $e'$ (so $\lambda^* = 0$), and we are left only with the feature divergence term. Thus, even if $d(P_e(G'), P_{e'}(G'))$ were as low as $d(P_e(G_c), P_{e'}(G_c))$ on the surface, the true shift relevant to classification is larger for $G'$ due to the label-distribution change. In summary, excluding part of $G_c$ makes the cross-environment difference strictly worse in terms of maintaining a stable predictor.

Case 3: $G'$ both includes $G_s$ and misses part of $G_c$. In this scenario $G'$ contains some spurious components and is also missing some causal components. By the arguments above, such a $G'$ will suffer from both a larger marginal distribution shift (due to the spurious parts varying across $e, e'$) and a label conditional shift (due to incomplete causal information), each of which increases the divergence between $P_e(G')$ and $P_{e'}(G')$. Therefore, this case trivially yields $\Delta(G') > \Delta(G_c)$ as well.

Combining the cases, we conclude that any alternative subgraph $G'$ that is not the full causal subgraph incurs a strictly greater distribution discrepancy between environments than $G_c$ does. The causal subgraph $G_c$ uniquely achieves the minimal cross-environment divergence by exactly capturing the invariant factors and nothing extra.

Moreover, by focusing on $G_c$, the learning algorithm sees an input distribution that is as invariant as possible across environments (Assumption 2 ensures minimal shift in $G_c$), and the label-generating

mechanism is completely stable (Assumption 1 ensures $Y$ depends only on $G_c$ in all environments). Consequently, both the feature distribution shift and the label conditional shift are minimized. Any deviation from $G_c$ either introduces additional feature shift (by including $G_s$) or label shift (by losing part of $G_c$), hence increasing the overall divergence. In formal terms, for any divergence measure $d$, $d(P_e(G_c), P_{e'}(G_c)) < d(P_e(G'), P_{e'}(G'))$ for all $G' \neq G_c$. This proves the claim $\Delta(G_c) < \Delta(G')$.

Finally, note that by Lemma 3, using $G_c$ also ensures the support of the target distribution is covered by the source distribution (no out-of-support surprise in new environments), which means the classifier can confidently generalize without encountering completely novel feature combinations. In contrast, a subgraph $G'$ containing $G_s$ might lead to out-of-support samples in a new environment (since $G_s$ can take unprecedented values), which is another manifestation of increased distribution shift and would break the classifier as for Lemma 3.

In conclusion, extracting the true causal subgraph $G_c$ yields the most invariant representation across environments, while minimizing reasonable measurements of distribution shift. Any other choice $G'$ either violates the invariant label relationship or introduces extra environment-dependent variation, thereby increasing the cross-environment divergence. This completes the proof that $G_c$ uniquely minimizes distributional disparity across environments of Theorem 1.

$\square$

## C.5 Proof of Theorem 2

*Proof.* Under Assumption 3, we require that for robust performance, the test inputs should not be completely novel relative to training. We argue that focusing on $G_c$ satisfies this requirement across environments, whereas including $G_s$ may violate it. Because $G_c$ is tightly related to $Y$, **all environments that share the same task (same $Y$ definition) are likely to exhibit $G_c$ patterns that are necessary to produce** $Y$. Even if the marginal distribution $P_e(G_c)$ shifts a bit (e.g., some $G_c$ patterns become more or less frequent), the set of possible $G_c$ values remains linked to the support of $Y$. Unless the new environment introduces an entirely new causal factor (which would effectively change the task definition and violate Assumption 1), $G_c$ in the new environment should fall within the realm of possibilities seen in training (perhaps with different probabilities). For example, if $G_c$ is a subgraph motif that causally triggers a certain label, any environment where that label can occur will contain that motif in those instances; it would not spontaneously create a completely different unseen motif to cause the same label, since $Y$ still comes from $f(G_c)$. This intuitive argument is backed by the idea that the causal mechanism $f(\cdot)$ is invariant – one cannot get a new output $Y$ without the appropriate $G_c$ input, so new environments cannot generate different valid $G_c$'s for the same $Y$ (they could only omit some or add irrelevant decoration via $G_s$). Therefore, we expect $\mathrm{Supp}(P_{e'}(G_c)) \subseteq \mathrm{Supp}(P_{\mathrm{train}}(G_c))$ (or at least a strong overlap), for any environment $e'$ that does not fundamentally alter the nature of the task. This fulfills the support overlap condition of Lemma 3 for $Z = G_c$. By that lemma, the classifier $h_\phi(\cdot)$ (which without loss of generality we take as the optimal invariant predictor $f(\cdot)$ or an approximation thereof) will perform equally well in environment $e'$ as it did in training, because it is operating on familiar ground. The risk in $e'$ can thus remain as low as the risk in training, i.e. performance is stable.

Conversely, if one uses a subgraph $G'$ that includes spurious elements, the new environment might present combinations of $G'$ that were never seen before. For instance, perhaps in training, a certain spurious pattern in $G'$ always coincided with a certain label (making the classifier think it was a useful feature), but in a new environment that pattern might appear with a different label or in a new context. The classifier, having learned a correlation, will mispredict because this input lies outside the training support for the joint $(G', Y)$ distribution (the model never learned the correct response to that scenario). In formal terms, $P_{e'}(G')$ may put mass on regions of the $G'$ space where $P_{\mathrm{train}}(G')$ had nearly zero mass (for example, motif with unseen basis before). Thus $\mathrm{Supp}(P_{e'}(G')) \not\subseteq \mathrm{Supp}(P_{\mathrm{train}}(G'))$. The violation of support overlap triggers exactly the failure mode highlighted in Lemma 3: the classifier $h_\phi(\cdot)$ is asked to extrapolate. If $h_\phi(\cdot)$ is a complex model (e.g. deep network), it might still output something for those novel inputs, but there is no guarantee it aligns with the true label – in fact it often will not, as it relies on the wrong features. This leads to performance drops or even arbitrarily bad predictions in the new environment.

Thus, using the causal subgraph $G_c$ ensures that the classifier is always seeing data within (or very near) the domain it was trained on (since what changes across $e$ is mostly the frequency of $G_c$ features,

not the support itself), guaranteeing stable performance. In contrast, using a non-causal subgraph means the classifier is likely to eventually step out of distribution, suffering from the classic OOD generalization failure. We conclude the proof for Theorem 2. □

# D  Additional Discussion of Empirical Examples on Distribution Shift and Representation Norms

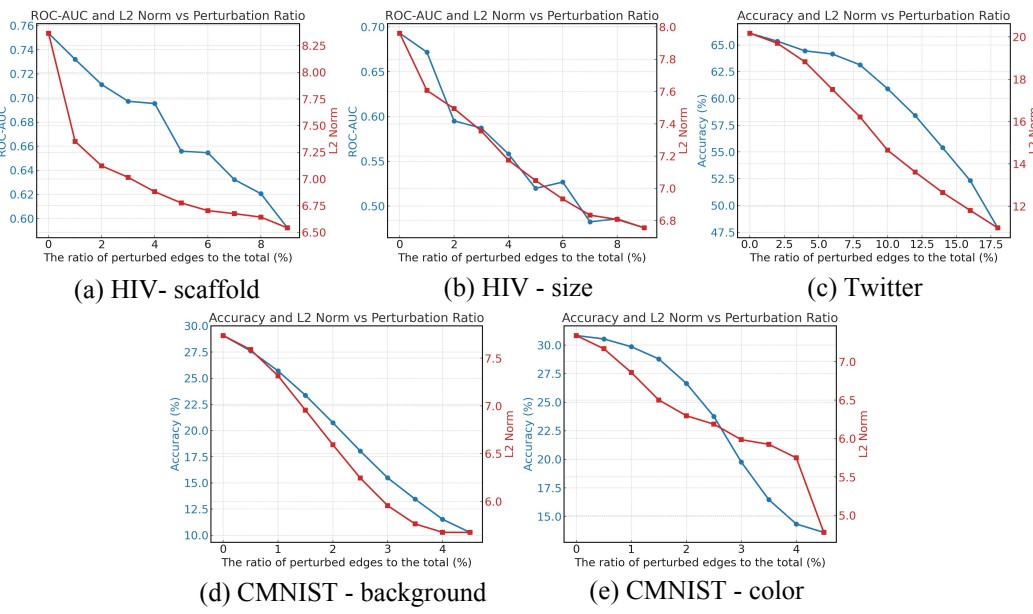

Figure 8: Structure causal models for graph data generation.

To illustrate that the representation norm decreases with increasing input distribution shift, we plot in Figure 2 (b) how the norm varies when we perturb the model's inputs. Figure 8 shows additional examples on other datasets. Specifically, following the experimental design described in the paper [16], we first train a high-accuracy GNN on the single-environment (no shift) dataset and freeze its parameters once converged. We then perturb the input graphs to simulate distribution shifts. Concretely, to model varying degrees of structural shift, we randomly insert a given proportion of edges into each input graph while simultaneously removing the same number of edges (thus preserving the total counts of nodes and edges so as to minimize any effects on the GNN's message-passing and aggregation). As the figures demonstrate, the GNN is highly sensitive to structural shifts: as the shift magnitude grows, the overlap between the perturbed inputs and the low-dimensional weight subspace diminishes, causing the representation norm to fall and the model's prediction accuracy to decline. These results show that the norm can be a robust indicator of distribution-shift severity.

# E  Low-Rankness in Graph Neural Networks

In neural networks, *low-rank* usually refers to the case where a layer's weight matrix can be approximated by a matrix with lower rank. This property is widely used in model compression, fast inference, and generalization analysis. A common method to evaluate low-rank is singular value decomposition (SVD). If most singular values are close to zero and only a few are large, the matrix is considered approximately low-rank.

We examine the *low-rank* of graph neural networks using two common models: Graph Convolutional Network (GCN) and Graph Isomorphism Network (GIN). Experiments are conducted on the synthetic dataset GOODMotif and the real-world dataset GOODSST2. Both models use three layers, and each GIN layer includes a two-layer MLP for feature transformation. As shown in the Figure 3 and 9, the weight matrices in each convolutional layer show clear low-rank patterns for both GCN and GIN.

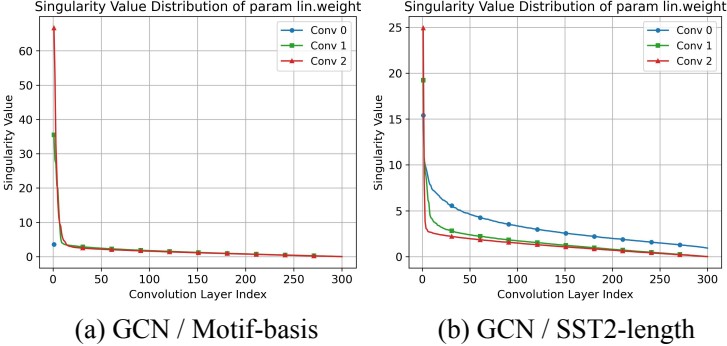

|  (a) GCN / Motif-basis | (b) GCN / SST2-length |

Figure 9: Singular value decomposition (SVD) results of GCN weights. Both models are trained with the empirical risk minimization (ERM) objective. Singular values are sorted in descending order. Clear low-rank patterns are observed across layers.

## F  A Toy Example of Low-Rank and Norm

Consider the weight matrix $W \in \mathbb{R}^{4 \times 3}$ given by:

$$W = \begin{pmatrix} 1 & 0 & 0 \\ 0 & 1 & 0 \\ 1 & 0 & 0 \\ 0 & 1 & 0 \end{pmatrix},$$

which has rank $\mathrm{rank}(W) = 2$. Now take two input vectors of unit length in different directions:

$$\mathbf{x}_1 = \begin{pmatrix} 1 \\ 0 \\ 0 \end{pmatrix}, \mathbf{x}_2 = \begin{pmatrix} 0 \\ 0 \\ 1 \end{pmatrix}$$

We compute their images under $W$, regardless of the bias:

$$W\mathbf{x}_1 = \begin{pmatrix} 1 \\ 0 \\ 1 \\ 0 \end{pmatrix}, \qquad W\mathbf{x}_2 = \begin{pmatrix} 0 \\ 0 \\ 0 \\ 0 \end{pmatrix}.$$

Consequently, their Euclidean norms satisfy

$$\|W\mathbf{x}_1\|_2 = \sqrt{1^2 + 0^2 + 1^2 + 0^2} = \sqrt{2} \quad > \quad \|W\mathbf{x}_2\|_2 = 0.$$

This simple example illustrates that a low-rank weight matrix can produce substantially different output norms for inputs aligned with its row-space versus those not aligned.

## G  Related Work

Out-of-distribution (OOD) generalization is a critical challenge in graph machine learning, as models trained on a given data distribution often fail to perform well on unseen distributions. Invariant learning, grounded in causal theory [33], is a primary approach to this problem: it seeks to learn causally relevant representations that remain stable across different environments. To acquire environment information, some methods leverage dataset-provided environment labels, e.g., IRM [1] and LECI [9], while others predict environment labels via unsupervised clustering, as in MoleOOD [46], GIL [22], and OOD-GCL [21], which entails prior assumptions about the environment distribution. Approaches such as DIR [44], GREA [29] and iMoLD [51] identify invariant features through structure- or feature-level disentanglement and recombination; CIGA [5], EQuAD [47], and LIRS [49] use self-supervised learning to separate invariant from spurious features.

Beyond invariant learning, alternative strategies have been developed to enhance generalization. DANN [7] applies domain-generalization techniques to tackle OOD issues; GSAT [32] and GOODGAT [43] exploit the graph information bottleneck to discover causal subgraphs; G-Splice [27] uses linear extrapolation to broaden dataset distributions; DGAT [10] leverages GAT [42]'s attention mechanism to strengthen GNN generalization; DIVE [40] makes predictions and summaries by selecting different and non-overlapping subgraphs from a single input graph respectively.

# H Datasets

We adopt two widely used benchmarks for graph OOD generalization—GraphOOD [8] and DrugOOD [15]—which together cover synthetic graphs, superpixel graphs, molecular graphs, and textual graphs:

• **GraphOOD:** a systematic benchmark tailored to graph OOD problems. We draw on four dataset groups of covarite shift in GraphOOD for graph classification: (1) **GOOD-Motif**: a synthetic dataset with two domain types—base-graph structure and graph size. (2) **GOOD-CMNIST**: a multi-class, semi-synthetic dataset obtained by converting Colored MNIST [1] into superpixel graphs, with different digit-color as domains. (3) **GOOD-HIV**: a real-world binary classification task predicting whether a molecule inhibits HIV replication, with scaffold and size as domains. (4) **GOOD-SST2** and **GOOD-Twitter**: sentiment-analysis tasks (binary and ternary, respectively) derived by encoding sentences as syntax trees, using sequence length as the domain.

• **DrugOOD**, an molecule OOD benchmark for drug discovery, defines three domain splits—assay, scaffold, and size—applied to two binding-affinity measurements (IC50 and EC50). This yields six binary-classification datasets, each predicting drug–target binding affinity.

As in prior work, we partition each dataset by its domain attribute to induce distribution shifts. For example, in the Motif basis-shift setting, the motif types in the test set are entirely disjoint from those in the training and validation sets, thus rigorously assessing model generalization.

We use the ROC-AUC metric for the binary classification dataset and Accuracy for the others. More details on the datasets can be found in the original papers [8, 15].

# I Baselines Details

We adopt the following methods as baselines for comparison:

**General methods:**

• **ERM** minimizes the empirical loss on the training set.

• **IRM** [1] seeks to find data representations across all environments by penalizing feature distributions that have different optimal classifiers.

• **Coral** [39] encourages feature distributions consistent by penalizing differences in the means and covariances of feature distributions for each domain.

• **VREx** [19] reduces the risk variances of training environments to achieve both covariate robustness and invariant prediction.

**Graph-specific OOD methods:**

• **DIR** [44] discovers the subset of a graph as invariant rationale by conducting interventional data augmentation to create multiple distributions.

• **GIL** [22] employs unsupervised clustering to infer environmental labels and leverages the invariant principle to identify causal subgraphs.

• **GSAT** [32] proposes to build an interpretable graph learning method through the attention mechanism and inject stochasticity into the attention to select label-relevant subgraphs.

• **CIGA** [5] proposes an information-theoretic objective to extract the desired invariant subgraphs from the lens of causality.

• **LECI** [9] assume the availability of environment labels, and study environment exploitation strategies for graph OOD generalization.

• **iMoLD** [51] employ environment augmentation techniques to facilitate the learning of invariant graph-level representations.

• **EQuAD** [47] adopts self-supervised learning to learn spuriosu efatures first, followed by learning invariant features by unlearning spurious features.

• **LIRS** [49] takes an indirect approach by first learning the spurious features and then removing them from the ERM-learned features.

Our selected baselines encompass a diverse array of approaches for tackling graph out-of-distribution (OOD) problems, including state-of-the-art and recently proposed methods. Some approaches such as OOD-GCL [21], GOODGAT [43], G-Splice [27] DGAT [10], et al. are omitted owing to a lack of comparable performance results or available implementation details. Moreover, the baselines we selected already encompass the main research directions of most state-of-the-art graph OOD methods.

## J   Visualized Cases

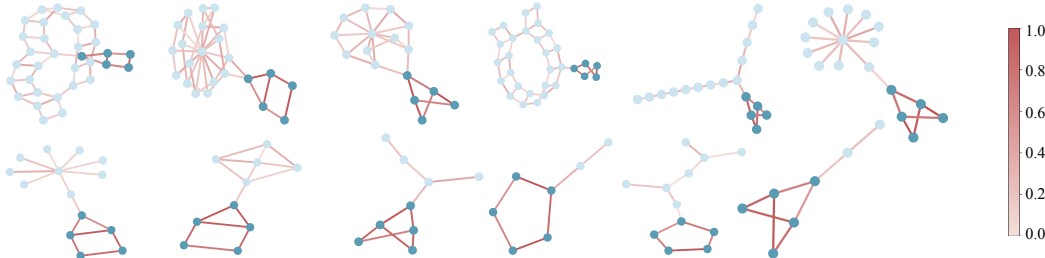

Figure 10: Visualized cases of the Motif-OOD dataset with size and basis domain shift. Nodes with dark blue and light blue colors represent the motif nodes and base graph nodes, respectively. The shading of the edges indicates the importance score of each edge generated by the subgraph extractor.

We present several visualized cases with size and basis domain shift in Figure 10. These examples demonstrate that our method can effectively extract the causal subgraph (Motif) from the input graph rather than selecting spurious factors (basis graphs). This also highlights the inherent interpretability of our approach.

## K   Discussion on the Effect of Top Ratio Hyperparameter

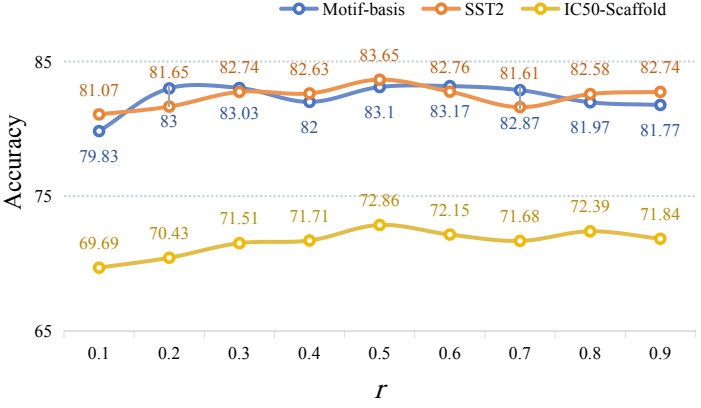

Figure 11: Performance on three datasets with different $r$.

In our approach, the hyperparameter $r$ controls the fraction of edges designated as the causal subgraph. However, the real edge ratio of subgraph in real-world datasets is typically unknown. As shown in Figure 11, we evaluate model accuracy across three datasets for different $r$ values and find that $r$ has no significant impact on generalization performance. From these results, we draw two empirical conclusions:

**Avoid overly small $r$:** If $r$ is set too low, the selected subgraph fails to fully cover the causal structure, degrading performance.

**Robustness for a moderate $r$:** When $r$ exceeds a minimal threshold, its exact value has little effect. We find that this robustness stems from the entropy-regularization term $\mathcal{L}_{comp}$ in 10, which automatically enforces an appropriate level of mask sparsity: some selected edges acquire near-zero mask weights and are thus effectively omitted during prediction.

Accordingly, we recommend $r = 0.5$ as a reasonable default for training.

## L  Efficiency Study

Experiments in this paper are conducted on NVIDIA RTX3090 GPUs. Our method is concise and streamlined: norm computation introduces virtually no additional overhead, and the two-stage parameter updates have a negligible impact on efficiency. Moreover, the Table 5 and 6 reports training and inference times of IDG and baselines, underscoring the high efficiency of our approach.

| Dataset | Training Batch Size | Testing Batch Size | Training Time (s) | Inference Time (s) |
|---------|---------------------|--------------------|--------------------|---------------------|
| ERM | 64 | 256 | 1032 | 1.3 |
| LECI | 64 | 256 | 3404 | 1.6 |
| DIR | 64 | 256 | 3213 | 1.7 |
| IDG | 64 | 256 | 1603 | 1.6 |

Table 5: Efficiency study of our method on Motif-basis.

| Dataset | Training Batch Size | Testing Batch Size | Training Time (s) | Inference Time (s) |
|---------|---------------------|--------------------|--------------------|---------------------|
| ERM | 64 | 256 | 1148 | 0.51 |
| LECI | 64 | 256 | 3973 | 1.43 |
| DIR | 64 | 256 | 3472 | 1.62 |
| IDG | 64 | 256 | 1855 | 1.53 |

Table 6: Efficiency study of our method on SST2.

## M  Broader Impact

Our work aims to enhance the generalization of graph neural networks (GNNs) in out-of-distribution (OOD) scenarios, which is crucial for real-world applications. By focusing on causal subgraph extraction, we provide a method that can potentially improve the robustness and reliability of GNNs in various domains, including drug discovery, social network analysis, and biological data interpretation. However, it is important to acknowledge that our approach may not be universally applicable to all graph-based tasks or datasets. The evaluations of our approach are mainly across limited graph domains, which may not represent all possible real-world scenarios. The approach can be evaluated on various environmental domains to be validated in a more realistic setting.

## N  Limitations

As with other graph generalization methods, although our approach improves the model's out-of-distribution performance to some extent, its transferability to other domains remains uncertain. Moreover, for datasets without any ground truth, leveraging the extracted subgraphs to further enhance generalization is a direction that has yet to be fully explored.

