# OpenReview forum: "Quantifying Distributional Invariance in Causal Subgraph for IRM-Free Graph Generalization"
_NeurIPS.cc/2025/Conference — NeurIPS 2025 poster_

### Official Review · Reviewer_Ti4B · 2025-07-02

**Clarity:** 4
**Significance:** 4
**Originality:** 3
**Rating:** 5
**Confidence:** 4

**Summary:**

This paper proposes a norm-guided invariant objective function for **quantifying distribution shift** in graph learning. The key insight is that representation norm correlates with distributional invariance of subgraphs, and can be used as a proxy signal to guide causal subgraph extraction, without relying on explicit environment labels. The method is theoretically motivated and empirically validated, and the proposed IDG framework is shown to perform well on multiple OOD benchmarks.

**Questions:**

1. Beyond Edge Perturbation: Can You Handle Real-World Shifts?  In the experiments (Section 2.3.1), the authors simulate OOD shift using edge perturbation. But in real-world datasets like **DrugOOD**, shifts come from **many sources,** like assay types, scaffold structures, dataset size, or even **activity cliffs**, where similar molecules can have very different labels. So I wonder: Does the norm-based quantification also work for these more complex or subtle shifts? Or is it mainly effective for simple structural changes?

2. About Continuous IRM. The quantitative method reminds me of **continuous IRM** methods that avoid hard environment partitions, like CORAL or DIR, and we can simply apply the soft assignment from multi-distribution (Dirichlet) to learn the continuous environments.     Could you kindly explain how your norm-based approach is similar or different from those in terms of objective and learning behavior?

3. Could Environment Modeling Help the Method? Even though your method is IRM-free, I don't think it conflicts with using environment info. For example, what would happen if we added known environment labels to your setup? Would the norm objective still behave the same, or would it change depending on how environments are defined? Maybe this could help reveal how sensitive your method is to the granularity of domain shifts.

**Ethical Concerns:**

["NO or VERY MINOR ethics concerns only"]

**Final Justification:**

Thank the authors for their hard work on the rebuttal. Most of the points are addressed, and I will maintain my score.

**Quality:**

3

**Strengths And Weaknesses:**

Strength:

1. The idea of quantifying OOD shift through representation norm is highly promising. This research direction has important implications for understanding model robustness and moving beyond heuristic environment partitioning in complicated out-of-distribution scenarios, e.g., drug discovery.
2. The paper connects the norm of representations with distributional shift, both theoretically (via Lemmas and Theorems) and empirically. This dual support makes the argument more convincing.
3. The writing is clear and easy to follow, and the paper is well-organized.

Weakness:
1. I had a hard time fully understanding Section 2.3.2. Could the authors please provide a simple example or an intuitive explanation of this part? Especially how the norm relates to empirical risk across domains.

2. The framework is IRM-free and avoids environment labels, which I appreciate. But the two-step training schedule (first freeze extractor, then predictor, and vice versa) looks similar to methods like environment-inference-based invariant learning methods (EIIL, GOODHSE, and MoleOOD, etc), which also extract latent structure before enforcing invariance.  What is the essential difference between alternating optimization and those used in environment inference-based methods?


Extended Questions (Personal Curiosity)
I’m very interested in this direction, so I also have some follow-up thoughts and questions in the Question part.

---

> ### Author Rebuttal · Authors · 2025-07-31
>
> Thank you for your appreciation on our work!
>
> ***Weakness***
>
> - **"Intuitive explanation of Section 2.3.2"** To better illustrate the connection between cross-environment/domain risk and the norm, we present the examples below:
>
>   1. Consider the weight matrix $W\in\mathbb{R}^{4\times3}$ given by:
>   $$W =
>   \begin{pmatrix}
>   (1 & 0 & 0),
>   (0 & 1 & 0),
>   (1 & 0 & 0),
>   (0 & 1 & 0)
>   \end{pmatrix}$$
>   which has rank $\mathrm{rank}(W)=2$.
>
>   1. We now pick two 'environmental' inputs $[C1,S1]$ from environment 1 and $[C2,S2]$ from environment 2, each decomposed into a causal part C and a spurious part S. From Theorem 1 that causal subgraph minimizes distribution shift across environments, we suppose $\mathrm{sim}(S_1,S_2)>\mathrm{sim}(C_1,C_2)$. Then we define:
>       - Env 1:
>       $$C_1 = \begin{pmatrix}1,0,0\end{pmatrix},\quad
>       S_1 = \begin{pmatrix}0,1,0\end{pmatrix}.$$
>       - Env 2:
>       $$C_2 = \begin{pmatrix}0.9,0.1,0\end{pmatrix},\quad
>       S_2 = \begin{pmatrix}0,0.1,0\end{pmatrix}.$$
>       Here $\mathrm{sim}(C_1,C_2) = 0.9$ while $\mathrm{sim}(S_1,S_2) = 0.1$, which satisfies $\mathrm{sim}(S_1,S_2)>\mathrm{sim}(C_1,C_2)$.
>
>   1. Ignoring the bias, we then consider the norm of the causal outputs after the model of weight $W$:
>   $$W\,C_1
>   = \begin{pmatrix}1,0,1,0\end{pmatrix},\quad W\,C_2
>   = \begin{pmatrix}0.9,0.1,0.9,0.1\end{pmatrix},\quad$$
>   $$\|W\,C_1\|_2 = \sqrt2 \approx 1.4142,$$
>   $$\|W\,C_2\|_2
>   = \sqrt{0.9^2 + 0.1^2 + 0.9^2 + 0.1^2}
>   = \sqrt{1.64}
>   \approx 1.2806.$$
>
>   1. The norm of the non-causal outputs after the model of weight $W$:
>   $$W\,S_1
>   = \begin{pmatrix}0,1,0,1\end{pmatrix},\quad
>   \|W\,S_1\|_2 = \sqrt2 \approx 1.4142,$$
>   $$W\,S_2
>   = \begin{pmatrix}0,0.1,0,0.1\end{pmatrix},\quad
>   \|W\,S_2\|_2 = \sqrt{0.1^2 + 0.1^2} = \sqrt{0.02} \approx 0.1414.$$
>
>   1. We can get that:
>   $$\mathbb{E}(\|W\,C\|_2) \approx 1.34 > \mathbb{E}(\|W\,C\|_2) \approx 0.78,$$ justifying the relationship between norm and distribution shift.
>
>   We will refine the description of the relationship in Section 2.3.2 and replace the example in Appendix F with this one to make a better explanation.
>
> - **"Compared to environment-inference-based invariant learning methods"** To clarify the distinction between the alternating optimization in our IDG method and environment inference approaches (e.g., EIIL, GOODHSE, MoleOOD), the following points are highlighted:
> 	1. In IDG training, alternating optimization is used mainly to prevent the subgraph extractor and predictor from converging to degenerate solutions or experiencing gradient conflicts when trained jointly under the same objective (classification loss and norm regularization). The extractor and predictor parameters are still updated alternately within each epoch, so they still train in a "collaborative" manner rather than being fully "decoupled" (for example, training the extractor to convergence before the predictor would be completely unworkable). Moreover, no explicit environment splits or pseudo-environment labels are produced.
> 	2. In environment inference approaches (e.g., EIIL, GOODHSE, MoleOOD), alternating optimization is required because the invariant learning phase demands explicit environment partitions or labels. These methods first infer environment information and then carry out invariant learning or related strategies. This two-stage process—environment inference followed by invariant learning—should be strictlly followed and can be fully decoupled, as in EIIL and MoleOOD.
>
>   We also conduct experiments on DrugOOD to compare performance under alternating versus non-alternating optimization. The results below show that alternating optimization significantly outperforms non-alternating optimization.
>
>   |DrugOOD|IC50Scaffold|IC50Size|IC50Assay|EC50Scaffold|EC50Size|EC50Assay|
>   |-|-|-|-|-|-|-|
>   |ERM|68.79|67.50|71.63|64.98|65.10|67.39|
>   |IRM|67.22|61.58|71.15|63.86|59.19|67.77|
>   |MoleOOD|68.02|66.51|71.38|66.69|65.09|73.25|
>   |GOODHSE|69.43|68.64|73.26|69.15|66.87|80.15|
>   |**IDG**|**69.97**|**69.02**|**73.34**|**69.57**|**68.03**|**80.54**|
>   |**IDG-noalter**|**67.97**|**67.23**|**70.84**|**65.43**|**65.67**|**71.24**|
>
>   Thank you for your kind attention to the training procedure. We will include comparisons with the environment-inference-based invariant learning methods you mentioned—such as GOODHSE and MoleOOD—in both the Related Work and Experimental sections in the future version.
>
> ***Questions***
>
> - **"Handle Real-World Shifts"** Indeed, on real-world datasets our method can also mitigate distribution shifts to some extent. The results on the DrugOOD benchmark, as you mentioned, are presented in the table above in the response of weakness 2. More importantly, we observe that our assumption—that causal subgraph distribution shifts are relatively small—still holds across multiple real-world datasets. Because real-world datasets lack ground-truth annotations for causal subgraphs, validating this hypothesis on them is extremely challenging. Nevertheless, we devised two distinct experimental setups to validate our assumption under different settings:
>
>   - To further support the validity of the causal subgraphs being more invariant in real applications, we conducte two experiments on real world datasets comparing causal and non-causal subgraphs. Quantifying distribution shifts in real-world graphs is challenging, since such data with ground-truth annotation of causal subgraph are rarely available, so we devise two distinct experimental setups to validate our assumption under different settings. Distribution shift was measured by the maximum mean discrepancy (MMD) between representations produced by the a same GNN model, using a Gaussian kernel on standardized features with bandwidth set by the median heuristic. We denote the cross‐environment distribution shift of causal and non‐causal subgraph by $\Delta{G_c}\approx MMD(G_c^{e1},G_c^{e2}), \Delta{G_s}\approx MMD(G_s^{e1},G_s^{e2})$ and compute the ratio of $\Delta{G_s}/\Delta{G_c}$. Table values above 1 indicate that non‐causal subgraphs undergo larger distribution shifts across environments. The experimental designs and results are described below:
>
>     - Experiment 1. Four real-world molecular graph datasets with ground-truth causal subgraphs—Mutag, Benzene, Fluoride, and Alkane- from the graph explanation literature. Each dataset was split into two environments by molecular size or random. The results are below:
>
>     |$\Delta{G_s}/\Delta{G_c}$|Mutag||Benzene||Fluoride||Alkane||
>     |-|-|-|-|-|-|-|-|-|
>     |split|size|random|size|random|size|random|size|random|
>     |MMD|3.43|1.56|5.79|1.83|6.67|1.94|6.48|2.03|
>
>     The results indicate that, the non-causal subgraph exhibits a larger shift than the causal subgraph across all environment partitions($\Delta{G_s}>\Delta{G_c}$), which aligns with Assumption 2.
>
>     - Experiment 2. We apply the same evaluation to the out-of-distribution (OOD) dataset used in our study. Due to the absence of true ground truth, we treat the subgraphs extracted by our method, GIL, and GSAT as causal subgraphs. The results are below:
>
>     |$\Delta{G_s}/\Delta{G_c}$|HIV-scaffold|HIV-size|SST2|Twitter|IC50-size|IC50-scaffold|
>     |-|-|-|-|-|-|-|
>     |IDG|7.28|3.915|1.963|3.983|1.1784|1.5296|
>     |GIL|1.969|1.429|1.044|3.25|1.154|1.74|
>     |GSAT|5.292|4.671|1.868|1.244|1.128|1.156|
>
>     The results indicate that, the extracted non-causal subgraph by different methods still exhibits a larger shift than the causal, which also aligns with Assumption 2.
>
>   These two experiments, viewed from different perspectives, clarify the validity of Assumption 2—that causal subgraphs typically exhibit smaller distribution shifts across environments.
>
> - **"Continuous IRM"** Indeed, our norm can be viewed as a continuous indicator of environments, but unlike continuous IRM, our method does not rely on any **hard or soft partitioning** of environments. It only assumes that the dataset contains data from different environments. Regarding the continuous IRM you mentioned, such as CORAL and DIR align the full-batch feature distributions—either in a latent or explicitly defined environment space—and enforce a single predictor to perform consistently across all environments. In contrast, our approach does not group or softly assign environments, nor does it explicitly align batch-level environmental distributions. Instead, we maximize a norm-based objective to identify at the sample-level causal subgraphs that are least prone to drift, without any hard or soft environment partitioning.
>
> - **"Environment Modeling Help the Method?"** Yes our method is fully compatible with IRM. When explicit environment labels are available, shifts in non-causal features across environments can be more evident. Furthermore, we integrat the IRM objective into our framework. As shown in the table below, this integration yields improved performance.
>   | Method   | PIIF  | FIIF  | SP=0.6 | SP=0.7 | SP=0.8 | SP=0.9 |
>   |----------|-------|-------|--------|--------|--------|--------|
>   | ERM| 62.45 | 37.22 | 91.67  | 91.25  | 90.63  | 85.92  |
>   | IRM| 68.34 | 44.33 | 91.85  | 91.25  | 90.48  | 85.38  |
>   | IDG| 79    | 83.67 | 92.67  | 92.06  | 91.89  | 91.33  |
>   | **IDG+ERM**  | **80.52**| **84.63** | **92.67**  | **92.34**  | **92.00**  | **91.89**  |
>
>   In this table, SP denotes the probability of spurious correlation. Lower SP values indicate weakly informative environments. PIIF and FIIF stand for partially informative and fully informative scenarios, respectively. These datasets adhere to the GALA[2], CIGA[3], and LECI[1] settings.
>
> [1] Joint Learning of Label and Environment Causal Independence for Graph Out-of-Distribution Generalization
>
> [2] Does Invariant Graph Learning via Environment Augmentation Learn Invariance?
>
> [3] Learning Causally Invariant Representations for Out-of-Distribution Generalization on Graphs

---

### Official Review · Reviewer_9Fyu · 2025-07-03

**Clarity:** 3
**Significance:** 3
**Originality:** 2
**Rating:** 4
**Confidence:** 4

**Summary:**

This paper proposes an IRM-free approach for out-of-distribution (OOD) generalization in graph neural networks. The core idea is that causal subgraphs tend to exhibit smaller distributional shifts across environments compared to non-causal parts. The authors formalize this observation as the Invariant Distribution Criterion and provide supporting theoretical analysis. They further establish a quantitative relationship between distribution shift and representation norm, using it to design a norm-guided objective for identifying causal subgraphs without requiring explicit environment labels. Experimental results on multiple OOD graph benchmarks show that the proposed method outperforms various strong baselines.

**Questions:**

How sensitive is the method to cases where the causal subgraph does not have a significantly larger norm than spurious parts, or when distributional shift affects the causal structure itself? Has the method been tested in partial information or weakly informative environments where spurious correlations are subtle rather than dominant?

**Ethical Concerns:**

["NO or VERY MINOR ethics concerns only"]

**Final Justification:**

Most of my concerns are addressed only except for related work discussions.

**Limitations:**

Yes

**Quality:**

2

**Strengths And Weaknesses:**

The paper provides a formal theoretical framework that links causal subgraph stability with distribution invariance. It introduces a simple and interpretable mechanism—representation norm—as a proxy for detecting distributional shifts. The method avoids explicit environment annotations, which enhances its practical applicability. The experimental evaluation is conducted on diverse datasets and includes ablation studies and comparisons with many competitive methods.

However, the key insight that causal subgraphs are more invariant is derived from intuition and synthetic examples, and its generalizability to real-world datasets is not rigorously validated. The theoretical results depend on strong assumptions that may not hold in practice, and the paper lacks a detailed discussion of these limitations. The idea of leveraging norm as a proxy for distribution shift has appeared in prior work in other domains, and the contribution here appears incremental in the absence of deeper justification for why this mechanism specifically works for graph data. Related work on invariant learning and graph causal reasoning is not thoroughly contrasted.

---

> ### Author Rebuttal · Authors · 2025-07-31
>
> Thank you for the feedback and please find below responses to your comments.
>
> ***Weakness***
>
> - **"Generalizability to real-world"** Because real-world datasets lack ground-truth annotations for causal subgraphs, validating this hypothesis on them is extremely challenging. Nevertheless, we devise two distinct experimental setups to validate our assumption under different settings:
>
>   - To validate that causal subgraphs are more invariant, we ran two experiments on real-world graphs, comparing causal and non-causal subgraphs under different settings. Given the scarcity of ground-truth causal annotations, we design distinct experimental setups and quantify distribution shift via maximum mean discrepancy (MMD) between GNN representations using a Gaussian kernel on standardized features with bandwidth set by the median heuristic. We denote the cross‐environment distribution shift of causal and non‐causal subgraph by $\Delta{G_c}\approx MMD(G_c^{e1},G_c^{e2}), \Delta{G_s}\approx MMD(G_s^{e1},G_s^{e2})$ and compute the ratio of $\Delta{G_s}/\Delta{G_c}$. Table values above 1 indicate that non‐causal subgraphs undergo larger distribution shifts across environments. The experimental designs and results are described below:
>
>     - Experiment 1. Four real-world molecular graph datasets with ground-truth causal subgraphs—Mutag, Benzene, Fluoride, and Alkane- from the graph explanation literature. Each dataset was split into two environments by molecular size or random. The results are below:
>
>     |$\Delta{G_s}/\Delta{G_c}$|Mutag||Benzene||Fluoride||Alkane||
>     |-|-|-|-|-|-|-|-|-|
>     |split|size|random|size|random|size|random|size|random|
>     |MMD|3.43|1.56|5.79|1.83|6.67|1.94|6.48|2.03|
>
>     The results indicate that, the non-causal subgraph exhibits a larger shift than the causal subgraph across all environment partitions($\Delta{G_s}>\Delta{G_c}$), which aligns with Assumption 2.
>
>     - Experiment 2. We apply the same evaluation to the out-of-distribution (OOD) dataset used in our study. Due to the absence of true ground truth, we treat the subgraphs extracted by our method, GIL, and GSAT as causal subgraphs. The results are below:
>
>     |$\Delta{G_s}/\Delta{G_c}$|HIV-scaffold|HIV-size|SST2|Twitter|IC50-size|IC50-scaffold|
>     |-|-|-|-|-|-|-|
>     |IDG|7.28|3.915|1.963|3.983|1.1784|1.5296|
>     |GIL|1.969|1.429|1.044|3.25|1.154|1.74|
>     |GSAT|5.292|4.671|1.868|1.244|1.128|1.156|
>
>     The results indicate that, the extracted non-causal subgraph by different methods still exhibits a larger shift than the causal, which also aligns with Assumption 2.
>
>   We will include these explanations and experiments in Section 2.2 (Theoretical Insights) of the paper.
>
> - **"Limitations about the assumption"** We discuss the rationale, limitations, and extreme violation cases here:
>
>   - *Assumption 1 (Causal Mechanism)* The generation of causal labels is fully determined by the causal subgraph $G_c$. This assumption derives from the Independent Causal Mechanism (ICM) hypothesis [6] and underpins Graph OOD methods such as CIGA[7] and LECI[5]. Invariant Risk Minimization (IRM) likewise relies on it by seeking representations that yield the same optimal classifier across environments. For instance, knowing the causal subgraph (e.g., functional groups) enables inferring a molecule's properties in different environments.
>
>   - *Assumption 2 (Environmental Diversity and Interventions)* This assumption originates from Peters et al.'s ICP framework and its extensions [1-4], this assumption holds that distributional shifts across environments arise from sparse interventions on the non-causal subgraph $G_s$, while $G_c$ remains stable. This hypothesis has been adopted by Graph OOD methods such as LECI[5], which impose an even stronger assumption—that the environment $E$ is independent of $G_c$. i.e. $E \perp G_c$. For instance, functional groups remain consistent in type and structure, while non-causal elements (e.g., heteroatoms) vary across environments.
>
>   - *Assumption 3 (Effective Classifier Support)* This assumption is based on the classical domain-adaptation bounds presented first in [8], this assumption requires overlap between the support of source and target distributions to guarantee generalization. Since classification relies on $G_c$, the support of the causal-subgraph distribution in the test domain must lie within that of the training domain. For example, a classifier can only correctly label a functional group at test time seen during training.
>
>   Limitations and Extreme Violations:
>   - *Assumption 1:* If causal labels depend on factors beyond G_c, our model may miss key causal relationships, harming classification. For example, if labels reflect post-reaction metrics rather than intrinsic chemical properties. **This scenario is at odds with existing work[1-9].** Like other Graph OOD and causal inference studies, our study focus solely on input–output mapping.
>   - *Assumption 2:* If shifts involve global feature changes rather than sparse interventions on $G_s$, this assumption fails—for instance, if environments are defined by critical functional groups, but **these cases are extremely rare** and we cannot find supported datasets and evidence from current Graph OOD benchmarks.
>   - *Assumption 3:* If the support of $G_c$ in the test domain does not overlap with that of the training domain—e.g., when test samples include unseen functional groups—no existing method can classify effectively. **This scenario also lies outside the scope of Graph OOD tasks in general.**
>
>   We will include the discussion about explanations and limitations of 3 assumptions in Section 2.2 in the paper.
>
> - **"Incremental contribution of leveraging norm"** The work [10] discussed in our paper, to the best of our knowledge, is the only work we have found to be related to us. However, compared to [10], our work differs in three key aspects:
>
>   - *Target application.* While [10] examines whether inputs are from different distributions, our method focuses on training models to correctly classify inputs from different distribution.
>   - *Evaluation domain.* Work [10] is conducted entirely within the vision domain, whereas our study is dedicated to graph data and graph neural networks.
>   - *Theoretical foundation.* The conclusions in [10] are based on analyses of linear networks, while our framework is grounded in causal inference, domain generalization, and graph learning.
>
>   Moreover, our contributions are threefold:
>     - We are the first to empirically establish a quantitative relationship between distribution shift and representation norms in graph data, and to explain its connection to the low-rank property of GNNs. (As recognized by Reviewer Ti4B)
>     - We provide a rigorous theoretical proof that causal subgraph minimizes distribution shift across environments. (As recognized by Reviewer Ti4B, rEir)
>     - We are the first to introduce norms into graph OOD research and to offer a theoretical justification for using norms to identify causal subgraphs. (As recognized by Reviewer q4r2, Ti4B)
>
> - **"Related work on invariant learning and causal reasoning"** Due to space constraints, details of existing invariant-learning and causal-reasoning baselines are provided in Appendices G and I. Please refer to our response to reviewer rEir *(Weak1: Optimization framework)* and Ti4B *(Weak2: Compared to environment-inference-based methods)*. and we plan to add a more in-depth discussion of how our method differs from these approaches in the Related Work section in the paper.
>
> ***Questions***
>
> - **"Cases where the causal subgraph does not have a larger norm/distributional shift affects the causal"** These settings contradict the assumptions and scenarios of both our work and prior Graph OOD studies like in [5,9,7,11]. In our experiments, we did not encounter any cases in which the causal subgraph fails to exhibit a larger norm than the spurious components, nor cases in which distributional shift affected the causal structure. If possible, could you suggest a dataset or benchmark that exhibits these conditions? We would then use the recommended benchmark to present further experimental results.
>
> - **Testing in partial information or weakly informative environments** We conduct experiments in these datasets. The results below demonstrate that our method performs strongly across all scenarios (denoting the variant without the norm objective as IDG-0):
>
>   ||SP=0.1|SP=0.3|SP=0.6|SP=0.7|SP=0.8|SP=0.9|PIIF|FIIF|
>   |-|-|-|-|-|-|-|-|-|
>   |ERM|94.53|93.26|91.67|91.25|90.63|85.92|62.45|37.22|
>   |IRM|93.65|92.87|91.85|91.25|90.48|85.38|68.34|44.33|
>   |GIL|92.14|91.57|89.78|80.33|72.00|51.09|59.66|64.66|
>   |DIR|90.63|90.10|88.70|87.82|86.00|84.80|69.33|42.00|
>   |**IDG**|**95.30**|**94.67**|**92.67**|**92.06**|**91.89**|**91.33**|**79.00**|**83.67**|
>   |IDG-0|94.89|94.29|90.43|90.16|86.74|83.13|66.33|70.66|
>
>   SP denotes the probability of spurious correlation, with lower values indicating weakly informative environments. PIIF and FIIF refer to partially and fully informative scenarios. These datasets adhere to the settings in [5,7,9]. We will incorporate these results and their analysis into the comparative experiments in Section 4.2.
>
> [1]Causal inference using invariant prediction: identification and confidence intervals
>
> [2]Invariant Causal Prediction for Nonlinear Models
>
> [3]Invariant Causal Prediction for Sequential Data
>
> [4]Invariance, Causality and Robustness
>
> [5]Joint Learning of Label and Environment Causal Independence for Graph Out-of-Distribution Generalization
>
> [6]Elements of Causal Inference: Foundations and Learning Algorithms
>
> [7]Learning Causally Invariant Representations for Out-of-Distribution Generalization on Graphs
>
> [8]A theory of learning from different domains
>
> [9]Does Invariant Graph Learning via Environment Augmentation Learn Invariance?
>
> [10] Deep Neural Networks Tend To Extrapolate Predictably
>
> [11] GOOD: A Graph Out-of-Distribution Benchmark

---

> > ### Comment · Reviewer_9Fyu · 2025-08-04
> >
> > Thank the authors for the responses. I still have concerns on the discussions on related works (invariant learning on graphs, debiased learning). I will keep my score unchanged.

---

> > > ### Author Response · Authors · 2025-08-06
> > >
> > > Thank you for your response. Here I would like to offer more detail on the invariant learning, causal reasoning, and debiased learning you mentioned.
> > >
> > > Invariant learning, which enforces causal invariance across environments in the learned representations, is grounded in causality/causal‐reasoning theory.  As a result, invariant learning and causal reasoning are widely used in graph OOD generalization tasks and should generally be inseparable.  Moreover, in the context of graph learning, they **share several important commonalities**:
> > >
> > > - Pursuit of invariance: Invariant learning emphasizes that the predictor or representation remain unchanged across multiple environments, while in graph causal reasoning the causal mechanism itself demands that the mapping from a causal subgraph $G_c$ to the label $Y$ remain invariant under interventions or distribution shifts.
> > > - Removal of spurious correlations: Both aims to strip away environment‐dependent noise features and retain only those signals with genuine causal explanatory power.
> > > - Complementary methodologies: Many existing invariant‐learning methods build directly on tools from graph causal inference (e.g., SCM assumptions, d‐separation, back‐door/front‐door criteria) or even treat the invariant subgraph as the causal subgraph itself.
> > >
> > > By contrast, debiased learning[1] and OOD generalization (including invariant learning and graph causal reasoning) **differ in key respects**:
> > > - Problem definition: In debiased learning, the biasing factor is typically known and accompanied by per‐sample metadata. While in graph OOD generalization, it does not assume knowledge of which specific bias is at work but only multiple (unknown) environments or domains induce distributional shifts.
> > > - Core objective: Debiased learning focuses on 'removal' of known, labeled bias features to prevent the model from relying on them, whereas graph OOD methods center on 'stability'—identifying invariant, truly causal subgraphs or embeddings that will generalize across arbitrary, unseen distribution changes.
> > >
> > > Here, we present **additional works** that our paper may not cover, as well debiased learning (could seldom studied on graphs and **of little relevance to OOD generalization**).
> > >
> > > - Graph OOD on invariant learning:
> > >   GOODHSE[2] hierarchically generates semantic graph environments via stochastic subgraph extraction and a contrastive objective to boost invariant OOD generalization, while MoleOOD[3] unsupervisedly infers latent environments and enforces environment-invariant substructure learning through a two-stage training objective for robust molecular representations under distribution shifts.
> > > - Graph OOD not on invariant learning:
> > >   AIA[4] aims to extrapolate and generate new environments, while concurrently preserving the original stable features during the augmentation process.
> > >   GALA[5] leverages an assistant model to detect distribution shifts in terms of such variance and maximally learns the causal information based on proxy predictions.
> > >   DGCL[6] learns disentangled representations via factor-wise contrastive self-supervision, avoiding the heavy cost of graph reconstruction.
> > >   IDGCL[7] improves DGCL by introducing independence regularization to remove dependencies among multiple disentangled representations.
> > > - Graph OOD related to debiased learning:
> > >   DisC[8] pioneers graph‐level debiased learning by using a differentiable neural GLCM to extract texture bias and a parameter‐free HEX module to remove it, yielding domain‐invariant representations without target‐domain data.
> > >
> > > Concerns and discussion about graph causal learning and invariant learning **have been covered in the original submission in Section 4&5** (Related Work&Experiments) **and Appendix G&I** (Related Work Details and Baselines Details), where we systematically present the **implementation philosophies, methodologies and empirical comparisons**.
> > >
> > > If there are **any related works we missed** or if you can share **more details(e.g. methods&benchmarks) about your specific concern**, please feel free to point them out, and we will gladly provide further clarification. Due to character limits, I can only provide this much. You may refer to [9] for more works on graph OOD.
> > >
> > > [1]Learning not to learn: Training deep neural networks with biased data.
> > >
> > > [2]Improving Out-of-Distribution Generalization in Graphs via Hierarchical Semantic Environments
> > >
> > > [3]Learning Substructure Invariance for Out-of-Distribution Molecular Representations
> > >
> > > [4]Unleashing the power of graph data augmentation on covariate distribution shift
> > >
> > > [5]Does invariant graph learning via environment augmentation learn invariance?
> > >
> > > [6]Disentangled contrastive learning on graphs
> > >
> > > [7]Disentangled graph contrastive learning with independence promotion
> > >
> > > [8]Debiasing Graph Neural Networks via Learning Disentangled Causal Substructure
> > >
> > > [9]A Survey of Deep Graph Learning under  Distribution Shifts: from Graph Out-of-Distribution  Generalization to Adaptation

---

### Official Review · Reviewer_q4r2 · 2025-07-03

**Clarity:** 3
**Significance:** 2
**Originality:** 3
**Rating:** 4
**Confidence:** 2

**Summary:**

This paper aims to address out-of-distribution generalization in graph neural networks. The main contribution of the paper is the development of an IRM-free method, i.e., the invariant distribution criterion, for capturing causal subgraphs, which are shown to exhibit smaller distributional variations across different environments compared to non-causal components. Specifically, they propose a norm-guided invariant distribution objective to efficiently discover and predict causal subgraphs, demonstrating superior performance over state-of-the-art methods in extensive experiments.

**Questions:**

Please refer to my summary of weakness.

**Ethical Concerns:**

["NO or VERY MINOR ethics concerns only"]

**Final Justification:**

I have carefully read the rebuttal provided by the authors and confirm that they have addressed my concerns. Therefore I maintain my positive score.

**Limitations:**

Yes.

**Paper Formatting Concerns:**

No.

**Quality:**

3

**Strengths And Weaknesses:**

This paper has the following advantage that I appreciate:

+ The paper introduces a novel method for graph generalization that does not rely on the traditional IRM framework, making it more practical and cost-effective.

+ Extensive experiments on multiple benchmarks seem to show that the proposed method consistently outperforms existing state-of-the-art methods.

However, the paper also have the following weakness:

- My main question lies in the practicality of the assumption. The core theory presumes distribution shifts affect only non-causal parts more than causal ones (Assumption 2). In real applications, causal mechanisms can drift too, potentially breaking guarantees.

- Lack in-depth discussion of the limitations. I'm curious that since representation norm can be influenced by multiple other effects, how confidence are we to attribute the norm difference to the distributional shift due to spurious correlations?

---

> ### Author Rebuttal · Authors · 2025-07-31
>
> Thank you for your appreciation on our work!
>
> ***Weakness***
>
> - **"Assumption 2, distribution shifts affect only non-causal parts"** This assumption originates from Peters et al.'s ICP framework and its extensions [1–4], this assumption holds that distributional shifts **across environments** arise from sparse interventions on the non-causal subgraph $G_s$, while $G_c$ remains stable. This hypothesis has been adopted by Graph OOD methods such as LECI[5], which impose an even stronger assumption—that the environment $E$ is independent of $G_c$. i.e. $E \perp G_c$. For instance, the types and structures of functional groups in a molecule are limited and stable, whereas non-causal parts (e.g., heteroatoms) vary widely across environments.
>
>   - To further support the validity of Assumption 2 in real applications, we conducted two experiments on real world datasets comparing causal and non-causal subgraphs. Quantifying distribution shifts in real-world graphs is challenging, since such data with ground-truth annotation of causal subgraph are rarely available, so we devised two distinct experimental setups to validate our assumption under different settings. Distribution shift was measured by the maximum mean discrepancy (MMD) between representations produced by the a same GNN model, using a Gaussian kernel on standardized features with bandwidth set by the median heuristic. We denote the cross‐environment distribution shift of causal and non‐causal subgraph by $\Delta{G_c}\approx MMD(G_c^{e1},G_c^{e2}), \Delta{G_s}\approx MMD(G_s^{e1},G_s^{e2})$ and compute the ratio of $\Delta{G_s}/\Delta{G_c}$. The experimental designs and results are described below:
>
>     - Experiment 1. Four real-world molecular graph datasets with ground-truth causal subgraphs—Mutag, Benzene, Fluoride, and Alkane—were selected from the graph explanation literature. Each dataset was split into two environments by molecular size and by random assignment. The results are presented below:
>
>     |$\Delta{G_s}/\Delta{G_c}$|Mutag||Benzene||Fluoride||Alkane||
>     |-|-|-|-|-|-|-|-|-|
>     |split|size|random|size|random|size|random|size|random|
>     |MMD|3.43|1.56|5.79|1.83|6.67|1.94|6.48|2.03|
>
>     The results indicate that, the non-causal subgraph exhibits a larger shift than the causal subgraph across all environment partitions($\Delta{G_s}>\Delta{G_c}$), which aligns with Assumption 2.
>
>     - Experiment 2. We apply the same evaluation to the out-of-distribution (OOD) dataset used in our study. Due to the absence of true ground truth, we treat the subgraphs extracted by our method, GIL, and GSAT as causal subgraphs. The results are presented below:
>
>     |$\Delta{G_s}/\Delta{G_c}$| HIV-scaffold | HIV-size | SST2  | Twitter | IC50-size| IC50-scaffold |
>     |-|-|-|-|-|-|-|
>     | IDG| 7.28| 3.915|1.963| 3.983| 1.1784 | 1.5296 |
>     | GIL | 1.969 |1.429| 1.044 | 3.25| 1.154| 1.74|
>     | GSAT  | 5.292| 4.671| 1.868 | 1.244 | 1.128| 1.156 |
>
>     The results indicate that, the extracted non-causal subgraph by different methods still exhibits a larger shift than the causal subgraph($\Delta{G_s}>\Delta{G_c}$), which also aligns with Assumption 2.
>
> - **"Norm difference to  the distributional shift due to spurious correlations"** We performe a detailed comparison between IDG and a variant without the norm objective (denoted IDG-0) across varying levels of spurious correlation strength to assess how much the norm objective contributes to mitigating spurious correlations. As shown in the table below, the norm objective's impact on performance grows with increasing spurious correlation strength, with a *Pearson correlation coefficient* of 0.86 between the spurious correlation probability and the performance gain. This finding demonstrates the effectiveness of our method in addressing spurious correlations.
>
>   |       | SP=0.5 | SP=0.6 | SP=0.7 | SP=0.8 | SP=0.9 | SP=0.95|
>   |-------|--------|--------|--------|--------|--------|--------|
>   | IDG   | 93.33  | 92.67  | 92.06  | 91.89  | 91.33  | 83.26  |
>   | IDG-0 | 90.66  | 90.43  | 90.16  | 86.74  | 83.13  | 70.72  |
>   |$\Delta$| 2.67  | 2.24   | 1.90   | 5.15   | 8.20   | 12.54  |
>
>   SP denotes the spurious-correlation probability. We will incorporate these results and their analysis into the comparative experiments in Section 4.2.
>
> [1] Causal inference using invariant prediction: identification and confidence intervals
>
> [2] Invariant Causal Prediction for Nonlinear Models
>
> [3] Invariant Causal Prediction for Sequential Data
>
> [4] Invariance, Causality and Robustness
>
> [5] Joint Learning of Label and Environment Causal Independence for Graph Out-of-Distribution Generalization

---

> ### Author Response · Authors · 2025-08-05
>
> Dear reviewer q4r2,
>
> Firstly, we would like to express our profound gratitude for your previous suggestions. Your recommendations were meticulous and constructive, prompting deeper reflections on many issues. We truly value the effort and thought you have invested in our work.
>
> We truly appreciate the time and expertise you invested in evaluating our manuscript. In response to your insightful comments, we have prepared detailed replies and implemented the suggested revisions to address each of your concerns.
>
> We sincerely hope to further discuss with you and ensure that we have completely met your expectations. If any aspects of our work remain unclear, or if further concerns arise, please do not hesitate to share them. Our commitment remains steadfast in providing clarity and responding to all feedback.
>
> Best regards,
>
> The Authors.

---

### Official Review · Reviewer_rEir · 2025-07-03

**Clarity:** 3
**Significance:** 3
**Originality:** 3
**Rating:** 4
**Confidence:** 4

**Summary:**

The paper proposes a method to extract causal subgraphs to address the problem of Out-of-Distribution (OOD) for graph classification. Their analysis unravels the relationship between distribution shifts and representation norms produced by GNNs: representation norms decay when distribution shifts increase. Thus, these findings inspire the paper’s methodology to extract causal subgraphs by incorporating the L2 representation norm into the loss function.

**Questions:**

Please refer to Weaknesses.

**Ethical Concerns:**

["NO or VERY MINOR ethics concerns only"]

**Final Justification:**

All of my concerns about the method's novelty and optimization details compared to previous Graph OOD works along with the method's time complexity details are fully addressed, so I decided to keep my rating score.

**Limitations:**

Please refer to Weaknesses.

**Paper Formatting Concerns:**

I do not have any concerns.

**Quality:**

3

**Strengths And Weaknesses:**

**Strength**:
* The paper’s organization is clear and easy to read.
* Propose detailed findings and theoretical analysis to inspire the methodology.
* Detailed report for experimental settings and reproducibility.

**Weaknesses**:
* Compared to previous works on extracting causal / invariant subgraphs (DIR [1], GIL [2]), IDG’s extractor architecture shares some similarities with these works’ extractor (node representation $\to$ masking probabilities $\to$ selecting top edges). Can you provide theoretical analysis or elaboration on why your proposed optimization framework (incorporating L2 norm) can help yield better performance than DIR [1]’s intervention mechanism or GIL [2]’s optimization framework?
* Can you provide time complexity analysis for IDG?
* Theoretical findings for IDG are backed by 3 assumptions listed in Section 2.2. I think it would make the paper more comprehensive if you include a brief explanation for why these 3 assumptions are reasonable and whether these 3 assumptions are also utilized by previous OOD works.

[1] Wu et al., Discovering Invariant Rationales for Graph Neural Networks. ICLR 2022.

[2] Li et al., Learning Invariant Graph Representations for Out-of-Distribution Generalization. NeurIPS 2022.

---

> ### Author Rebuttal · Authors · 2025-07-31
>
> Thank you for your appreciation on our work!
>
> ***Weakness***
>
> - **"Optimization framework comparing to DIR/GIL"** The extractor architecture employed here, which maps node features to edge masks, is widely used in Graph OOD research. Besides GIL and DIR, it is also adopted by methods such as LECI[1], GALA[2], CIGA[3], DiSC[4], and GSAT[5]. Its origins trace back to earlier graph explanation methods like PGExplainer[6], as well as similar techniques in the text[7] and vision domains[8]. We would like to point out that the key contribution of this work is a **new theoretical framework** that, without relying on invariant learning or any environmental labels, guarantees the extractor identifies causal subgraphs rather than the extracting procedure.  The main challenge of this framework lies in finding an effective optimization approach that jointly trains the extractor and classifier to improve their coordination on extracting causal features and predicting. Compared to DIR and GIL, our method offers the following key distinctions and advantages:
>
>   - 1, Both DIR and GIL build on invariant risk minimization (IRM) [9] by attempting to find environment-invariant predictors, which requires inferring each sample's environment label. DIR induces different distributions via explicit intervention perturbations, while GIL derives environment labels through unsupervised clustering of k-means. Neither approach has theoretical guarantees that the constructed environments match the real data-generating distributions, which is also noted in related work such as GALA[2] and CIGA[3]. Our experiments below further reveal two points:
>
>     - Removing the invariance (variance) objective from DIR and GIL (denoted DIR-0 and GIL-0) can actually improve performance in some settings.
>     - In certain scenarios, both DIR and GIL perform even worse than standard ERM.
>
>     |Dataset|PIIF|FIIF|SP=0.6|SP=0.7|SP=0.8|SP=0.9|
>     |-|-|-|-|-|-|-|
>     |ERM|62.45|37.22|91.67|91.25|90.63|85.92|
>     |IRM|68.34|44.33|91.85|91.25|90.48|85.38|
>     |GIL|59.66|64.66|89.78|80.33|72|51.09|
>     |GIL-0|53.67|37.49|90.78|78.45|79|83.46|
>     |DIR|69.33|42|88.7|87.82|86|84.8|
>     |DIR-0|53.67|36|87.67|89.79|87.8|86.74|
>     |**IDG**|**79**|**83.67**|**92.67**|**92.06**|**91.89**|**91.33**|
>
>     These datasets adhere to the GALA[2], CIGA[3], and LECI[1] settings. PIIF and FIIF are two typical distribution-shift scenarios, and SP denotes the spurious-correlation probability. **These results indicate that the optimization strategies of DIR and GIL are not universally effective.** In contrast, our method is completely **IRM-free and requires no environment information**: it identifies causal subgraphs simply by finding solutions that satisfy the minimal shift criterion via norm optimization.
>
>   - 2, From an optimization perspective, DIR also demands a large memory bank for non-causal features $S$, which hinders its scalability to large datasets, and GIL must collect representations across the entire batch for clustering, both of which incur substantial resource overhead. By comparison, our approach only requires two forward passes and minimal additional computation of computing the norm, demonstrating huge efficiency advantages.
>
>   - We will include a detailed comparison against DIR, GIL, and other related methods in the Related Work and Discussion sections in the paper.
>
> - **"Time complexity analysis"** The time complexity of the IDG method is $O(m d + n d^2)$, where $n$, $m$, and $d$ denote the average number of nodes, edges, and feature dimensions per graph. Specifically:
>
>   - Message passing and node updates (per GNN layer) incur a cost of $O(m d + n d^2)$.
>   - Edge scoring and top-r selection can be completed in $O(m \log m)$ in the worst case.
>   - Norm regularization and compactness term require $O(m)$ time.
>
>   Thus, a single forward–backward pass on one graph remains $O(m d + n d^2)$. In our method, the norm calculation incurs minimal computational overhead (less than that of edge selection). To further demonstrate its efficiency, we measured both training time (per epoch) and inference time on the Motif, HIV, and SST2 datasets. The results are presented in the table below:
>   ||Motif-basis||HIV-scaffold||SST2-length||
>   |-|-|-|-|-|-|-|
>   |Time(ms)|Training(ms)|Inference(ms)|Training(ms)|Inference(ms)|Training(ms)|Inference(ms)|
>   |GIL|56801|2037|102594|22057|86658|16472|
>   |DIR|14415|794|38477|1098|32392|9377|
>   |**IDG**|**11262**|**493**|**35232**|**995**|**26326**|**7659**|
>
> - **"Explanation of 3 assumption"** We discuss the rationale of these three core assumptions here:
>
>   - *Assumption 1 (Causal Mechanism)*
> The generation of causal labels is fully determined by the causal subgraph $G_c$. This assumption derives from the Independent Causal Mechanism (ICM) hypothesis [9] and underpins Graph OOD methods such as CIGA[3] and LECI[1]. Invariant Risk Minimization (IRM) likewise relies on it by seeking representations that yield the same optimal classifier across environments. For example, if the causal subgraph (e.g., functional groups responsible for key chemical properties) is known, one can infer a molecule's chemical properties regardless of its distribution of origin (environments).
>
>   - *Assumption 2 (Environmental Diversity and Interventions)*
> This assumption originates from Peters et al.'s ICP framework and its extensions [10–13], this assumption holds that distributional shifts across environments arise from sparse interventions on the non-causal subgraph $G_s$, while $G_c$ remains stable. This hypothesis has been adopted by Graph OOD methods such as LECI, which impose an even stronger assumption—that the environment $E$ is independent of $G_c$. i.e. $E \perp G_c$. For instance, the types and structures of functional groups in a molecule are limited and stable, whereas non-causal parts (e.g., heteroatoms) vary widely across environments.
>
>   - *Assumption 3 (Effective Classifier Support)*
> This assumption is based on the classical domain-adaptation bounds presented first in [14], this assumption requires overlap between the support of source and target distributions to guarantee generalization. Since classification relies on $G_c$, the support of the causal-subgraph distribution in the test domain must lie within that of the training domain. For example, a classifier can correctly and consistently label a functional group in the test environment only if that group appeared during training.
>
>   - To further support the validity of Assumption 2 in real applications, we conducted two experiments on real world datasets comparing causal and non-causal subgraphs. We devised two distinct experimental setups to validate our assumption under different settings. Distribution shift was measured by the maximum mean discrepancy (MMD) between representations produced by the a same GNN model, using a Gaussian kernel on standardized features with bandwidth set by the median heuristic. We denote the cross‐environment distribution shift of causal and non‐causal subgraph by $\Delta{G_c}\approx MMD(G_c^{e1},G_c^{e2}), \Delta{G_s}\approx MMD(G_s^{e1},G_s^{e2})$ and compute the ratio of $\Delta{G_s}/\Delta{G_c}$. The experimental designs and results are described below:
>
>     - Experiment 1. Four real-world molecular graph datasets with ground-truth causal subgraphs—Mutag, Benzene, Fluoride, and Alkane—were selected from the graph explanation literature. Each dataset was split into two environments by molecular size and by random assignment. The results are presented below:
>
>     |$\Delta{G_s}/\Delta{G_c}$|Mutag||Benzene||Fluoride||Alkane||
>     |-|-|-|-|-|-|-|-|-|
>     |split|size|random|size|random|size|random|size|random|
>     |MMD|3.43|1.56|5.79|1.83|6.67|1.94|6.48|2.03|
>
>     The results indicate that, the non-causal subgraph exhibits a larger shift than the causal subgraph across all environment partitions($\Delta{G_s}>\Delta{G_c}$), which aligns with Assumption 2.
>
>     - Experiment 2. We apply the same evaluation to the out-of-distribution (OOD) dataset used in our study. Due to the absence of true ground truth, we treat the subgraphs extracted by our method, GIL, and GSAT as causal subgraphs. The results are presented below:
>
>     |$\Delta{G_s}/\Delta{G_c}$| HIV-scaffold | HIV-size | SST2  | Twitter | IC50-size| IC50-scaffold |
>     |-|-|-|-|-|-|-|
>     | IDG| 7.28| 3.915|1.963| 3.983| 1.1784 | 1.5296 |
>     | GIL | 1.969 |1.429| 1.044 | 3.25| 1.154| 1.74|
>     | GSAT  | 5.292| 4.671| 1.868 | 1.244 | 1.128| 1.156 |
>
>     The results indicate that, the extracted non-causal subgraph by different methods still exhibits a larger shift than the causal subgraph($\Delta{G_s}>\Delta{G_c}$), which also aligns with Assumption 2.
>
>   These two experiments, viewed from different perspectives, clarify the validity of Assumption 2. We will include these explanations and experiments in Section 2.2 (Theoretical Insights) of the paper.
>
>   We will include the explanations of 3 assumptions in Section 2.2 in the paper.
>
> [1] Joint Learning of Label and Environment Causal Independence for Graph Out-of-Distribution Generalization
>
> [2] Does Invariant Graph Learning via Environment Augmentation Learn Invariance?
>
> [3] Learning Causally Invariant Representations for Out-of-Distribution Generalization on Graphs
>
> [4] Debiasing graph neural networks via learning disentangled causal substructure
>
> [5] Interpretable and Generalizable Graph Learning via  Stochastic Attention Mechanism
>
> [6] Parameterized Explainer for Graph Neural Network
>
> [7] Rationalizing Neural Predictions
>
> [8] Interpreting Image Classifiers by Generating  Discrete Masks
>
> [9] Elements of Causal Inference: Foundations and Learning Algorithms
>
> [10] Causal inference using invariant prediction: identification and confidence intervals
>
> [11] Invariant Causal Prediction for Nonlinear Models
>
> [12] Invariant Causal Prediction for Sequential Data
>
> [13] Invariance, Causality and Robustness
>
> [14] A theory of learning from different domains

---

### Decision · Program_Chairs · 2025-09-17

**Decision:**

Accept (poster)

**Comment:**

This paper initially got mixed scores: one accept, one borderline reject and two borderline accept. The authors have submitted a rebuttal, and after considering it, all reviewers expressed satisfaction with the responses and updated their scores to borderline acceptance (score 4/5). The AC concurs with the reviewers that this submission has the advantages of good results, novel idea, and well-written. Therefore, the AC recommends acceptance and encourages the authors to incorporate the clarifications—particularly the more detailed discussion of related work and the additional experiments provided in the rebuttal—into the final version.